

# The use of biochar in animal feeding

Hans-Peter Schmidt[1], Nikolas Hagemann[1,2], Kathleen Draper[3] and
Claudia Kammann[4]

[1] Ithaka Institute for Carbon Strategies, Arbaz, Valais, Switzerland
[2] Environmental Analytics, Agroscope, Zurich, Switzerland
[3] Ithaka Institute for Carbon Intelligence, Victor, NY, USA
[4] Department of Applied Ecology, Hochschule Geisenheim University, Geisenheim, Germany

Corresponding author
Hans-Peter Schmidt,
schmidt@ithaka-institut.org

## ABSTRACT

Biochar, that is, carbonized biomass similar to charcoal, has been used in acute medical treatment of animals for many centuries. Since 2010, livestock farmers increasingly use biochar as a regular feed supplement to improve animal health, increase nutrient intake efficiency and thus productivity. As biochar gets enriched with nitrogen-rich organic compounds during the digestion process, the excreted biochar-manure becomes a more valuable organic fertilizer causing lower nutrient losses and greenhouse gas emissions during storage and soil application. Scientists only recently started to investigate the mechanisms of biochar in the different stages of animal digestion and thus most published results on biochar feeding are based so far on empirical studies. This review summarizes the state of knowledge up to the year 2019 by evaluating 112 relevant scientific publications on the topic to derive initial insights, discuss potential mechanisms behind observations and identify important knowledge gaps and future research needs. The literature analysis shows that in most studies and for all investigated farm animal species, positive effects on different parameters such as toxin adsorption, digestion, blood values, feed efficiency, meat quality and/or greenhouse gas emissions could be found when biochar was added to feed. A considerable number of studies provided statistically non-significant results, though tendencies were mostly positive. Rare negative effects were identified in regard to the immobilization of liposoluble feed ingredients (e.g., vitamin E or Carotenoids) which may limit long-term biochar feeding. We found that most of the studies did not systematically investigate biochar properties (which may vastly differ) and dosage, which is a major drawback for generalizing results. Our review demonstrates that the use of biochar as a feed additive has the potential to improve animal health, feed efficiency and livestock housing climate, to reduce nutrient losses and greenhouse gas emissions, and to increase the soil organic matter content and thus soil fertility when eventually applied to soil. In combination with other good practices, co-feeding of biochar may thus have the potential to improve the sustainability of animal husbandry. However, more systematic multi-disciplinary research is definitely needed to arrive at generalizable recommendations.

## INTRODUCTION

Biochar is produced by pyrolysis from various types of biomass in a low-to-no oxygen thermal process at temperatures ranging from 350 to 1,000 °C (*European Biochar Foundation (EBC), 2012*; *International Biochar Initiative (IBI), 2015*). Using water vapor or $CO_2$ at temperatures above 850 °C or chemical compounds like phosphoric acid and potassium chloride, the biochar undergoes an activation process resulting in activated biochar (i.e., activated carbon) (*Hagemann et al., 2018*). When produced from pure stem wood, the solid phase of the pyrogenic process is known as *charcoal*. In contrast, the term *biochar* indicates that a broad spectrum of biogenic materials can serve as feedstock. Biochar, activated carbon and charcoal can all be considered as pyrogenic carbon materials.

The term biochar indicates that it is used for any purpose that does not involve its rapid mineralization to $CO_2$ (e.g., burning it) (*European Biochar Foundation (EBC), 2012*). In a broader sense, the term *biochar* denotes its intended long-time residence in the terrestrial environment, either as a soil amendment or for other material-use purposes (*Schmidt et al., 2018*). Since biochar-carbon decomposes much slower than the original biomass, the application and use of biochar is considered as a terrestrial carbon sink on at least a centennial scale (*Zimmerman & Gao, 2013*; *Lehmann et al., 2015*; *Werner et al., 2018*) and is therefore a promising negative emission technology (*IPCC, 2018*).

During the first decade of modern biochar research summarized in *Lehmann & Joseph (2015)*, biochar was usually tested as a soil amendment that was applied pure to soils in large quantities (>10 t/ha) revealing modest to large yield increases for a multitude of crops in the tropics but only rarely in temperate climates (*Jeffery et al., 2017*). More recently it was (re-)discovered that blending biochar with organic amendments such as manure, cattle urine or compost may increase yields more significantly and in a broader spectrum of climates and soils (*Steiner et al., 2010*; *Kammann, Glaser & Schmidt, 2016*; *Godlewska et al., 2017*; *Schmidt et al., 2017*). As quality biochar is non-toxic and thus even feedable and edible (*European Biochar Foundation (EBC), 2012*), this apparently favorable combination of organic residues with biochar prompted researchers and a rapidly increasing number of practitioners to conduct trials where biochar was not only mixed with manure but also included as an input into animal farming systems. The incremental addition of biochar to silage, feed, bedding material and liquid manure pit demonstrated that biochar can be used in cascades. In addition to the direct benefits for animal husbandry as discussed below in detail, biochar becomes thus enhanced with organic nutrients which increases the economic viability of biochar application while providing numerous environmental benefits along the (cascading) way.

When combined with silage, biochar can reduce mycotoxin formation, bind pesticides, suppress butyric acid formation and enhance the quantity of lactic bacteria (*Calvelo Pereira et al., 2014*). Farmers observed that when biochar was combined with straw or saw dust bedding at 5–10% (vol) hoof diseases, odors and nutrient losses were reduced (*O'Toole et al., 2016*). Moreover, farmers reported that adding 0.1% biochar (m/m) in a liquid manure pit reduced odors, surface crust and nutrient losses (*Schmidt, 2014*; *Kammann et al., 2017*). Throughout these cascades, the biochar becomes enriched with

organic nutrients and functional groups, while the cation exchange capacity and redox activity increases, and pH decreases (*Joseph et al., 2013*). Analyses indicate that, by enriching the biochar with liquids organic nutrients (whether in the digestive tract, bedding, manure pit or by co-composting), the interior surfaces of the porous biochar become drenched with an organic coating (*Hagemann et al., 2017*; *Joseph et al., 2018*). This increases both water storage capacity and nutrient exchange capacity (*Conte et al., 2013*; *Kammann et al., 2015*; *Schmidt et al., 2015*). The biochar becomes thus a more efficient plant growth enhancing soil amendment, that improves the recycling of nutrients from organic residues of animal farming (*Kammann et al., 2015*). The cascading use of biochar in animal farming systems also reduces the environmentally harmful loss of ammonia through volatilization or nitrate through leaching (*Liu et al., 2018*; *Borchard et al., 2019*; *Sha et al., 2019*) and it has the potential to reduce greenhouse gas emissions such as nitrous oxide ($N_2O$) (*Kammann et al., 2017*; *Borchard et al., 2019*), or methane (CH4) (*Jeffery et al., 2016*). To the best of our knowledge, no study so far has quantified biochar emission reduction effects along a full cascade. The studies cited above are reviews or meta-analyses summarizing mainly effects of the amendment of biochar to soil.

When in 2012 the cascading use of biochar and especially its addition to animal feed began in Germany and Switzerland (*Gerlach & Schmidt, 2012*), the biochar market in Europe started to grow considerably. Since then, the largest proportion of industrially produced biochar in Europe is sold for animal feed, bedding, manure treatment and thus subsequent soil application (*Kammann et al., 2017*; *O'Toole et al., 2016*; *Schmidt & Shackley, 2016*). In 2016, the European Biochar Foundation introduced a new biochar certification standard specifically for animal feed (*European Biochar Foundation (EBC), 2018*) to allow for quality control, as well as conformity with European regulations for animal feed.

When ingested orally, biochar has been shown to improve the nutrient intake efficacy, adsorb toxins and to generally improve animal health (*O'Toole et al., 2016*; *Toth & Dou, 2016*). After numerous veterinary papers published last century, a number of scientific studies on biochar feeding have been published since 2010, dealing with biochars' impact on the health of various animal species, on feed efficiency, pathogen infestation and on greenhouse gas emissions. Thus, we review the current state of knowledge regarding the use of biochar as a animal feed additive. We identify systematic gaps in the scientific understanding as it is still mechanistically unclear why biochar, as a feed additive, causes the observed effects. We also highlight potential side effects, the known and potential effects on greenhouse gas emissions, the necessity for adapted regulatory practice and quality control as well as the need for dedicated research to close knowledge gaps.

## RESEARCH METHODS

This study predominantly selected research papers published between 1980 and 2019 but included also a selection of historical articles and books published between 1905 and 1979. Some rare oral communications were included to reference and illustrate farmer and feed certifier experiences.

## Search strategy

We searched the following electronic databases: Science Direct, Scopus, ISI Web of Science and Research Gate. To identify the relevant publications, we used the following search terms: (biochar OR charcoal OR activated carbon) and (animal OR feed OR livestock OR livestock type (cow, poultry, sheep etc.) OR methane OR pesticides OR silage OR manure). The references cited in the reviewed studies were also included in the search and scanned separately for relevant publications. To summarize the historical literature (20 studies) we used the Karlsruhe Virtual Catalogue and the literature cited in the respective historical works in English, German and French. We further interviewed Dr. Achim Gerlach, a veterinarian who has been treating large cattle herds with biochar for nearly a decade; only a small fraction of his experiences are published in peer-reviewed journals (*Gerlach & Schmidt, 2012*).

## Selection of studies

The authors assessed the titles and abstracts of all retrieved references of relevance to the objective of this review. Due to the relatively small number of studies, we included all studies that investigated biochar or charcoal or activated carbon in vivo as feed additive for improving performance and animal health (27 studies). We further selected in vivo or in vitro studies when animal tissue or digestive liquids were used as medium and if they were related to mycotoxin- (26 studies), bacteria related pathogen- (22 studies), poisoning and drug overdoses (21 studies), and pesticide- (23 studies) adsorption or methane emissions (12 studies). In total, 112 scientific studies on biochar effects in animal feeding were reviewed. Reported results were only discussed as significant when $p < 0.05$ was obtained in the respective study.

# RESULTS AND DISCUSSION

## Historical overview

### The use of biochar/charcoal as feed or feed additive before 2010

Charcoal is one of the oldest remedies for digestive disorders, not only for humans but also for livestock. Cato the Elder (234 -149 BC) was one of the first to mention it in his classic *On Agriculture*: "If you have reason to fear sickness, give the oxen before they get sick the following remedy: 3 grains of salt, 3 laurel leaves, [ . . . ], 3 pieces of charcoal, and 3 pints of wine." (*Cato, 1935*, §70). Besides the administration of medicinal herbs, oil or clay, charcoal was widely used by traditional farmers all over the world for internal disorders of any sort. Apparently, it never did any harm but was mostly beneficial (*Derlet & Albertson, 1986*). For some animals like chicken or pigs, the charcoal was administered pure; for others it was mixed with butter (cows), with eggs (dogs) or with meat (cats).

A textbook on animal husbandry dating from 1906 observed: "Swine appear to have a craving for what might be called 'unnatural substances'. This is especially true of hogs that are kept in confinement, which will eat greedily such substances as charcoal, ashes, mortar, soft coal, rotten wood etc. It is probable that some of the substances are not good for hogs, but there is no doubt that charcoal and wood ashes have a beneficial effect, the former being greatly relished" (*Day, 1906*).

19th century and early 20th century agricultural journals printed many discussions on the benefits of various "cow tonics," mostly composed of charcoal and a variety of other ingredients including spices, such as cayenne pepper, and digestive bitters like gentian. Manufacturers of these tonics claimed they would reduce digestive disorders, increase appetite and improve milk production (*Pennsylvania State College, 1905*).

At this time in the USA, charcoal was considered a superior feed additive for increasing butterfat content of milk. Cow's milk was tested for butterfat content in competitions where top-producing cows could win a prize. Farmers took great care in formulating the feed ration for such tests: *The grain mixture fed during the test consisted of 100 pound of distillers dried grains, 50 pounds of wheat bran, 100 pounds of ground oats, 100 pounds of hominy, 100 pounds of cottonseed meal . . . . Charcoal is seldom if ever left out the test ration by many of the breeders*" (*Savage, 1917*).

The use of activated and non-activated biochar feed for animal health was already being researched and recommended by German veterinarians at the beginning of the last century. Since 1915, research into activated biochar had revealed its effect in reducing and adsorbing pathogenic clostridial toxins from *Clostridium tetani* and *Clostridium botulinum* (*Skutetzky & Starkenstein, 1914*; *Luder, 1947*). *Mangold (1936)* presented a comprehensive study on the effects of biochar in feeding animals, concluding that *"the prophylactic and therapeutic effect of charcoal against diarrheal symptoms attributable to infections or to the type of feeding is known. In this sense, adding charcoal to the feed of young animals would seem a good preventive measure."* *Volkmann (1935)* described an effective reduction in excreted oocysts through adding biochar to the food of pets with coccidiosis or coccidial infections.

Later, *Totusek & Beeson (1953)* wrote that biochar products are used since at least 1880 in US-American hog breading and since 1940 in feed for poultry. In their influential article, the authors provided an extensive list of references. At around the same time, *Steinegger & Menzi (1955)* wrote: "*It is generally common in Switzerland to add biochar to chick feed and to the meal for laying hens to prevent digestive problems and to achieve a regulating effect on digestion.*"

### Biochar and wild animals

At first glance it might seem somewhat unnatural to feed biochar/charcoal to animals, but in fact even wild mammals occasionally eat biochar if it is available to them. In nature, charcoal residues from wild fires can still be found years later. Deer and elk are reported to eat from charred trees in Yellowstone National Park and domestic dogs to eat charcoal briquettes (*Struhsaker, Cooney & Siex, 1997*). The *Zanzibar red colobus* (*Procolobus kirkii*), a small monkey regularly eats charcoal to help digest young Indian Almond (*Terminalia catappa*) or mango (*Mangifera indica*) leaves that contain toxic phenolic compounds (*Cooney & Struhsaker, 1997*). *Struhsaker, Cooney & Siex (1997)* observed that individual colobus monkeys consumed about 0.25–2.5 g of charcoal per kg body weight daily. Additional adsorption tests performed by *Cooney & Struhsaker (1997)* indicated that in particular the African kiln charcoals (which the monkeys also ate) were surprisingly good at adsorbing hot-water-extracted organics from the

above-mentioned tree leaves. Thus, the authors concluded that the monkeys' charcoal consumption was likely a (self-)learned behavior, increasing the digestibility of their typical leaf diet. Interestingly, a population count of *colobus* monkeys on this African island showed that they reached the highest population density of all monkey species worldwide. It seems, therefore, that the daily consumption of such wood-based biochar has no negative long-term effect at least not on these monkeys.

## Mechanisms of biochar in feed digestion

### Adsorption

Before biochar was investigated and used as a regular feed additive for animals in the early 2010s, charcoal (i.e., biochar made from wood) and activated carbon (i.e., activated biochar when made from biomass; *Hagemann et al., 2018*) was considered a veterinary drug to tackle indigestion and poisoning. Charcoal was known for many centuries as an emergency treatment for poisoning in animals (*Decker & Corby, 1971*). Biochar has been and still is used because of its high adsorption capacity for a variety of different toxins like mycotoxins, plant toxins, pesticides as well as toxic metabolites or pathogens. Adsorption therapy, which uses activated biochar as a non-digestible sorbent, is considered one of the most important ways of preventing harmful or fatal effects of orally ingested toxins (*McKenzie, 1991*; *McLennan & Amos, 1989*).

From a toxicology perspective, most of the effects of biochar are based on one or several of the following mechanisms: selective adsorption of some toxins like dioxins, co-adsorption of toxin containing feed substances, adsorption followed by a chemical reaction that destroys the toxin and desorption of earlier adsorbed substances in later stages of digestion (*Gerlach & Schmidt, 2012*). However, classifiable distinctions need to be made to the time-dependent and partly overlapping processes of adsorption, biotransformation, desorption and excretion of the toxic substances throughout the digestive system of animals.

*Schirrmann (1984)* described the effects of activated carbon on bacteria and their toxins in the gastrointestinal tract as:

1. Adsorption of proteins, amines and amino-acids.
2. Adsorption of digestive tract enzymes, as well as adsorption of bacterial exoenzymes.
3. Binding, via chemotaxis, of mobile germs.
4. The selective colonization of biochar with gram-negative bacteria might result in decreased endotoxin release as these toxins could be directly adsorbed by the colonized biochar when gram-negative bacteria dying-off.

One further major advantage of the use of biochar is its "enteral dialysis" property, that is, already adsorbed lipophilic and hydrophilic toxins can be removed from the blood plasma by the biochar, as the adsorption power of the huge surface area of the biochar interacts with the permeability properties of the intestine (*Schirrmann, 1984*).

Susan *Pond (1986)* explained various mechanisms by which biochar can eliminate toxins from the body. First, biochar can interrupt the so-called enterohepatic circulation of

toxic substances between the intestine, liver and bile. It prevents compounds such as estrogens and progestagens, digitoxin, organic mercury, arsenic compounds and indomethacin from being taken up in bile. Second, compounds such as digoxin, which are actively secreted into the intestine, can be adsorbed there. Third, compounds such as pethidines can be adsorbed to the biochar, which passively diffuse into the intestine. Fourth, the biochar can take up compounds that diffuse along a concentration gradient between intestinal blood and primary urine.

### Redox activity of biochar-based feed additives

Although the adsorption capacity is the most prominent function of biochar to explain its positive impacts when fed to animals, adsorption alone cannot explain all phenomena that are observed in biochar feeding experiments. Another pivotal, but still widely overlooked function of biochar is its redox activity. Biochars act as so called *geobatteries and geoconductors* that can accept, store and mediate electrons from and for biochemical reactions (*Sun et al., 2017*). Low temperature biochars (HTT of 400–450 °C) function as geobatteries mainly due to their phenol and quinone surface groups. High temperature biochars (HTT >600°), on the other hand, are good electrical conductors (*Mochidzuki et al., 2003*; *Yu et al., 2015*). Due to both of these qualities, both, high and low temperature biochars, can act in biotic and abiotic redox-reactions as electron mediators (*Van Der Zee & Cervantes, 2009*; *Husson, 2012*; *Liu et al., 2012*; *Kappler et al., 2014*; *Kluepfel et al., 2014*; *Joseph et al., 2015a*; *Yu et al., 2015*; *Sun et al., 2017*). Biochar can accept and donate electrons as, for example, in microbial fuel cells where activated biochar can be used as an anode and as a cathode (*Gregory, Bond & Lovley, 2004*; *Nevin et al., 2010*; *Konsolakis et al., 2015*). The electrical conductivity of biochar is, however, not based on continuous electron flow, like in a copper wire, but on discontinuous electron hopping (*Kastening et al., 1997*), which is of essential importance for biochar's function as a (microbial) electron mediator or so-called electron shuttle, facilitating even inter-species electron transfer (*Chen et al., 2015*). Due to the comparably large size of biochar particles, the electron transfer capacity of biochar's carbon matrices may lead to a relatively long-distance electron exchange that provides a spatially more extensive accessibility to alternative electron acceptors such as minerals for anoxic microbial respiration (*Sun et al., 2017*).

During the microbial decomposition of organic substances in the gastrointestinal tract and particularly in the anaerobic rumen, digestive microbes require a terminal electron acceptor to get rid of surplus electrons that accumulate during the degradation of organic molecules. As electrons do not exist in a free state under ambient environmental conditions and cannot be stored in large enough quantities by cells, organisms always depend on the availability of both an electron donor (e.g., the metabolized organic matter) and an acceptor to which surcharge electrons can be transferred. This usually occurs in so-called redox reactions where molecules or atoms that donate an electron are coupled through electro-chemical reactions with molecules or atoms that accept an electron. To allow this electron transfer, these chemical or biochemical redox-reactions usually have to take place in very close (molecular) proximity.

The coupling of electron donating and electron accepting reactions can, however, be bridged by so-called electron mediators or electron shuttles. Those electron meditators can take up an electron from a chemical reacting molecule, solid interphase or microorganism and provide it to another molecule, atom, solid interphase or microorganism. Well known and investigated electron mediating compounds include thionine, tannins, methyl blue or quinone, showing comparable capacities to humic substances and biochar (*Van Der Zee et al., 2003*; *Liu et al., 2012*; *Bhatta et al., 2012*; *Kluepfel et al., 2014*).

A well-balanced animal feed regime should contain multiple electron mediating substances. In the high-energetic diets used in intensive livestock farming, the supply with electron-shuttling substances is, however, often insufficient (*Sophal et al., 2013*). When inert or other non-toxic electron mediators like biochar or humic substances are added to high-energy feed, several redox reactions may take place more efficiently, which could in turn increase the feed intake efficiency (*Liu et al., 2012*; *Leng, Inthapanya & Preston, 2013*). Biochar, specifically, can act as both a sole electron mediator or a synergistic electron mediator that increases the efficiency of other mediators (*Kappler et al., 2014*).

Inside the gastro-intestinal tract, nearly all feed-degrading reactions are facilitated by microorganisms (mostly bacteria, archaea and ciliates). Within those reactions, bacterial cells may transfer electrons to biofilms or via biofilms to other terminal electron acceptors (*Richter et al., 2009*; *Kracke, Vassilev & Krömer, 2015*). However, biofilms are rather poor electric conductors and the electron-accepting capacity is low. Hence, microbial redox reactions can be optimized by electron shuttles, such as humic acids or activated biochar whose electrical conductivity is 100–1,000 times higher than that of biofilms (*Aeschbacher et al., 2011*; *Liu et al., 2012*; *Saquing, Yu & Chiu, 2016*). Although the conductivity of non-activated biochar is lower compared to activated biochar, it has been shown that it can efficiently transfer electrons between bacterial cells (*Chen et al., 2015*; *Sun et al., 2017*). Bacteria were shown to donate an electron to a biochar particle while other bacteria of different species took up (accepted) an electron at another site of the same biochar particle. The biochar acts here like a "battery" (or electron buffer) that can be charged and discharged, depending on the need of biochemical (microbial) reactions (*Liu et al., 2012*). Moreover, as biochar can be temporarily oxidized or reduced by microbes (i.e., biochar is depleted or enriched in electrons), it can buffer situations with a (temporary) lack of electron donors or terminal electron acceptors (redox buffering effect) (*Saquing, Yu & Chiu, 2016*). A principal aim of feeding biochar to animals could thus be to overcome metabolic redox limitations by enhancing electron exchange between microbes, and between microbes and terminal electron acceptors.

The redox-active carbonaceous backbone of the biochar as well as minerals it contains, such as iron (Fe(II) and/or Fe(III)) and manganese (Mn(III) or Mn(IV) minerals), can electrically support microbial growth in at least four different ways: (1) as an electron sink for heterotrophy-based respiration, (2) as an electron sources for autotrophic growth, (3) by enabling cell-to-cell transfer of electrons and (4) as an electron storage material (*Shi et al., 2016*). It can be hypothesized that enabling of extracellular electron transfer contributes to a more energy efficient digestion resulting in higher feed efficiency when activated or non-activated biochar is administered. Moreover, the electrochemical

effects need to be considered as a major factor for explaining possible shifts in the functional diversity of the microbial community in the digestive system (*Prasai et al., 2016*). *Leng, Inthapanya & Preston (2012)* also suggested that electron transfer between biochar and microorganisms could be one of the reasons why feeding biochar to cows led to reduced methane emissions in their studies (see chapter 6).

It is further very likely that biochar has the function of a redox wheel in the digestive tract, comparable to $Fe^{III}$–$Fe^{II}$-redox wheels. It could act jointly as an electron acceptor and donator coupling directly various biotic and abiotic redox-reactions comparable to mixed valent iron minerals (*Davidson, Chorover & Dail, 2003*; *Li et al., 2012*; *Joseph et al., 2015a*; *Quin et al., 2015*). Beside its polyaromatic backbone, biochar contain, depending on the production process, a multitude of volatile organic carbons (VOC) (*Spokas et al., 2011*). Some of the pyrolytic VOCs are strong electron acceptors and may act, like a redox wheel similar to how quinone works (*Van Der Zee et al., 2003*). Some of these pyrolytic VOCs that often undergo oxidative modifications during the aging of biochar (*Cheng & Lehmann, 2009*) are so-called redox-active moieties (RAMs) that have been shown to contribute to the biodegradation of certain contaminants (*Yu et al., 2015*). It can be surmised that in the digestive tract, a multitude of RAMs, adsorbed on the surfaces of biochar particles, can act as redox-wheels with various microorganisms. It can be further hypothesized that when biochar buffers electrons in the vicinity of redox active surface groups, it may provide stabile micro-habitats with different redox-pH-milieus for different species of microorganisms (*Yu et al., 2015*). Moreover, biochar adsorbs certain feed and metabolic substances like tannins, phenols or thionin, which are also electron acceptors and which might further increase the electron buffering of biochar particles during its passage through the digestive tract (*Kracke, Vassilev & Krömer, 2015*).

Biochar, wood vinegar (i.e., aqueous solutions of condensed pyrolytic gases) and humic substances can act as redox buffering substances (*Husson, 2012*; *Kluepfel et al., 2014*) which may explain why the feeding of biochar, pyrolytic vinegar and humic substances often show similar effects; and why the blending of biochar with wood vinegar or humic substances seems to reinforce the effects (*Watarai, Tana & Koiwa, 2008*; *Gerlach et al., 2014*). However, unlike both dissolved organic substances, biochar provides a highly porous framework with high specific surface area, where humic-like substances or pyrolytic vinegar can be adsorbed and unfurl three-dimensionally as a coating of the inner-porous aromatic carbon surfaces of biochar. Due to the redox buffering effect of biochar blended with humic substances or wood vinegar, variations of the redox potential may be minimized in the proximity of biochar particles, which could support those species of microorganisms that find their optimum at these redox potentials (*Kalachniuk et al., 1978*; *Cord-Ruwisch, Seitz & Conrad, 1988*). Biochar particles may thus provide selective hotspots of microbial activity. It can be assumed that the buffering of the redox potential as well as the effect of electron shuttling between microbial species can have a selective, microbial milieu forming effect, which facilitates and accelerates the formation of functional microbial consortia (*Kalachniuk et al., 1978*; *Khodadad et al., 2011*; *Sun et al., 2017*).

The mechanistic understanding of biochar used as feed additive, especially with regard to its impact on microbial mediated redox reactions, is clearly in its infancy (*Gregory, Bond & Lovley, 2004*; *Nevin et al., 2010*; *Konsolakis et al., 2015*). However, we hypothesize with some confidence that biochar has a direct electro-chemical influence on digestive reactions, and that this is one, if not the main, reason for the extremely varying effects of different biochars. Electrical conductivity, redox potential, electron buffering (poising) and electron transfer capacity (shuttling) of a given biochar depend highly on the type of pyrolyzed feedstock, pyrolytic conditions (*Kluepfel et al., 2014*; *Yu et al., 2015*) and especially on pyrolysis temperature (*Sun et al., 2017*). The higher the temperature above 600 °C, the better is the electron transfer rate and electrical conductivity (*Sun et al., 2017*). However, the higher the VOC content of, for example, lower-temperature biochars and higher abundance of surface functional groups on lower temperature biochars (400–600 °C), the more important the mediated electron transfer onto/from the biochar may become (*Joseph et al., 2015a*; *Yu et al., 2015*; *Sun et al., 2017*). In addition, the mineral content of biochars should be taken into account as well, since it does not only influence biochar's electro-chemical behavior, but it may also catalyze various biotic and abiotic reactions (*Kastner et al., 2012*; *Anca-Couce et al., 2014*).

## Specific toxin adsorption

### Adsorption of mycotoxins

The contamination of animal feed with mycotoxins is a worldwide problem that affects up to 25% of the world's feed production (*Mézes, Balogh & Tóth, 2010*). Mycotoxins are mainly derived from mold fungi, whose growth on fresh and stored animal feed is difficult to prevent, especially in humid climates. Mycotoxin-contaminated feed can result in serious diseases of farm animals. To protect the animals, adsorbents are usually added to the feed to bind the mycotoxins before ingestion. In addition to the frequently used aluminosilicates, activated carbon and special polymers are increasingly being used (*Huwig et al., 2001*).

One of the most common mycotoxins is aflatoxin (*Alshannaq & Yu, 2017*), which has, therefore, been used in numerous studies as a model substance to investigate the adsorption behavior of biochar and how it reduces the uptake of the toxin in the digestive tract and hence in the animal blood and in milk (*Galvano et al., 1996a*). *Galvano et al. (1996b)* were able to reduce the extractable aflatoxin concentration in animal feed by up to 74% and the concentration in milk by up to 45%, by adding 2% activated biochar to pelleted aflatoxin-spiked feed for dairy cows. The non-systematic comparison of different activated biochars, however, showed that there are large differences in the adsorption efficiency between different types of (activated) biochar.

*Diaz et al. (2002)* showed in an in vitro sorption batch study that four different activated carbons adsorbed 99% of the aflatoxin B from a 0.5% aflatoxin B-spiked solution when activated biochars were dosed at 1.11 g on 100 ml. However, when Diaz administered 0.25% activated carbon to aflatoxin-B contaminated feed for dairy cows a year later (*Diaz et al., 2004*), they were unable to demonstrate any significant reduction in aflatoxin B levels in the milk. Here, it has to be considered that in the in vivo test, an insufficiently

characterized (activated) biochar was fed at a low concentration of 0.25% of the feed fresh weight, whereas in the in vitro studies, the biochar was added at 1% to the aqueous solution, that is, four times higher, and in the absence of a feed matrix.

*Galvano et al. (1996a)* also investigated the adsorption capacity of 19 different activated carbons for two mycotoxins, ochratoxin A and deoxynivalenol, and found that the activated biochar adsorbed 0.80–99.86% of the ochratoxin A and up to 98.93% of the deoxynivalenol, depending on the type of activated biochar. The large range of results clearly confirms the importance of a systematic characterization and classification of biochar properties. However, Galvano et al. concluded that neither the iodine number used for activated biochar characterization, nor the Brunauer–Emmet–Teller specific surface area derived from $N_2$ gas-adsorption isotherms allowed straightforward predictions of the adsorption capacity for these mycotoxins.

*Di Natale, Gallo & Nigro (2009)* compared various natural and synthetic adsorbent feed additives for dairy cows to reduce the aflatoxin content in milk. Activated biochar showed the highest toxin reduction capacity (>90% aflatoxin reduction in milk with 0.5 g aflatoxin per kg diet). Analytical studies of the milk quality also showed slight positive effects on the milk composition with regard to organic acids, lactose, chlorides, protein content and pH. The authors explained the high adsorption capacity with the high specific surface area in combination with a favorable micropore size distribution of the biochar, and the high affinity of aflatoxin for the polyaromatic surface of the biochar in general (*Di Natale, Gallo & Nigro, 2009*).

*Bueno et al. (2005)* investigated the adsorption capacity of various doses of activated biochar (0.1%, 0.25%, 0.5%, 1%) for zearalenone, a dangerous estrogenic metabolite of the fungus species Fusarium, for which so far no treatment agents had been found. In vitro, all zearalenone could be bound at each of the four biochar doses. However, in vivo, where a wide variety of mycotoxins and numerous other organic molecules compete with the free adsorption surfaces of biochar, hardly any specific adsorption could be achieved.

A study with Holstein dairy cows investigated to what extent the negative effects of fungal-contaminated feed silage can be reduced by co-feeding activated biochar at 0, 20 or 40 g daily (*Erickson, Whitehouse & Dunn, 2011*). Cows fed the biochar amendment and the contaminated silage had higher feed intake and improved digestibility of neutral detergent fiber, hemicellulose and crude protein and had higher milk fat content compared to the control without biochar. When the same daily amounts of biochar were administered to uncontaminated quality silage, no changes in digestion behavior, milk quality or any other effect on the dairy cows could be detected. However, the authors showed in a second experiment that cows, when given the choice, clearly preferred good quality silage to contaminated silage either with or without biochar. They concluded that farmers should focus on providing high quality feed rather than mitigating negative effects of contaminated silage with biochar.

While *Piva et al. (2005)* found no protection against the injurious effects of fumonisin, a highly toxic mycotoxin, following a 1% addition of biochar to the feed of piglets, Nageswara *Rao & Chopra (2001)* showed that the addition of biochar to aflatoxin B1 contaminated feed of goats reduced the transfer of the toxin (100 ppb) to the milk by 76%.

In the latter trial, the efficiency of activated biochar was significantly higher than that of bentonite (65.2%). Both adsorbents did not affect the composition of goat's milk nor the average level of milk production.

In vitro studies with porcine digestive fluids showed high rates of adsorption of *Fusarium* toxins such as deoxynivalenol (67%), zeralenone (100%) and nivalenol (21%) through activated biochar (*Avantaggiato, Solfrizzo & Visconti, 2005*; *Döll et al., 2007*). On the other hand, *Jarczyk, Bancewicz & Jedryczko (2008)* found no significant effect when they added 0.3% activated biochar to the diet of pigs. Neither in the blood serum nor in the kidneys, the liver or in the muscle tissue could the ochratoxin concentrations be reduced by this small amount of supplement with uncharacterized industrial biochar (*Jarczyk, Bancewicz & Jedryczko, 2008*). However, no adverse effect was noted either.

Mycotoxins often cause serious liver damage in poultry. Biochar administered at daily rates of 0.02% of the body weight significantly increased the activity of key liver enzymes (*Ademoyero & Dalvi, 1983*; *Dalvi & Ademoyero, 1984*). While aflatoxin (10 ppm) reduced feed intake and weight gain of broiler chickens, the addition of 0.1% biochar to the feed (w/w) reversed the negative trend (*Dalvi & McGowan, 1984*).

Comparing the effect of activated biochar with a conventionally used alumina product (hydrated sodium calcium aluminosilicate), it was found that the alumina product resulted in considerable liver and blood levels of aflatoxin B when administered at 0, 40, 80 μg AFB1 per kg diet, but not when combined with a 0.25% and 0.5% biochar treatment (*Kubena et al., 1990*; *Denli & Okan, 2007*). In another study, activated biochar reduced the concentration of aflatoxin B in the feces of chickens for fattening, but only if the biochar was administered separately from the feed (*Edrington et al., 1996*). However, *Kim et al. (2017)* showed with an in vivo pig feeding trial that the aflatoxin absorption capacity was reduced by 100%, 10% and 20%, respectively, for three different biochars supplemented at 0.5% to the same basal diet, again demonstrating the importance of considering specific biochar properties. The importance of dosage was confirmed in another recent poultry trial where 0.25% or 0.5% activated biochar was added to an aflatoxin B1 contaminated diet, decreasing aflatoxin B1 residues in the liver of the birds by 16–72%, depending on the aflatoxin B1 and biochar dosages (*Bhatti et al., 2018*).

In their review article, *Toth & Dou (2016)* document further conflicting studies in which biochar feeding may or may not mitigate the effects of mycotoxin intoxication. The results of most studies on sorption in aqueous solution (in vitro) did not correlate with the results in corresponding in vivo test results (*Huwig et al., 2001*). Thus, in vitro studies have to be interpreted with care, because matrix effects can dramatically impact mycotoxin sorption, for example, *Jaynes, Zartman & Hudnall (2007)* found that an activated carbon (Norit®, Boston, MA, USA) could sorb up to 200 g/kg aflatoxin, but only in clear solution. In a corn meal suspension, sorption capacity was 100 times lower due to matrix effects. Matrix effects in the digestive tract can be expected to be even more complex due to varying pH and redox conditions. Still, based on our review, we conclude that negative effects of certain mycotoxins such as deoxynivalenol (*Devreese et al., 2012*, *2014*; *Usman et al., 2015*) and zearalenone (*Avantaggiato, Havenaar & Visconti, 2004*) can be effectively suppressed with rather low dosages of activated biochar amended to feed, while no benefit

was found for aflatoxin. It can be hypothesized that (activated) biochar is only able to suppress negative effects of mycotoxins that are rather hydrophobic (*Avantaggiato, Havenaar & Visconti, 2004*).

However, most of these studies have in common that only commercial activated carbons and biochars were used without proper characterization, that is, systematic trials with biochar of different feedstock (e.g., wood vs. herbaceous feedstock) and production conditions (e.g., temperature) are barely available. Thus, systematization of the results remains difficult.

### Adsorption of bacteriological pathogens and their metabolites

The use of activated and non-activated charcoals to improve animal health was recommended and studied by German veterinarians as far back as the beginning of the 20th century. In 1914, the adsorbing effect of charcoal for various toxins in the digestive tract was described by *Skutetzky & Starkenstein (1914)*. First experiments with bacterial toxins of *Clostridium tetani* and *Clostridium botulinum* as well as with diphtheria toxin were performed as early as 1919 (*Jacoby, 1919*). In particular, Wiechowski pointed out how important the quality of the charcoal is, and how different the effect of different charcoals on the toxin adsorption can be (*Wiechowski, 1914*). Ernst Mangold described in 1936 the effect of charcoal in animal feeding comprehensively and concluded: "*The prophylactic and therapeutic effect of charcoal on infectious or feeding-related diarrhea is clear, and based on this observation, the co-feeding of charcoal to juvenile animals appears as an appropriate prevention*" (*Mangold, 1936*). At about the same time, Albert Volkmann published his findings about efficient reduction of oocyst excretion resulting from coccidiosis and coccidial infections when charcoal was fed to domestic animals (*Volkmann, 1935*).

*Gerlach et al. (2014)* demonstrated that daily supplement of 400 g of a high-temperature wood-based biochar (i.e., HTT 700 °C) significantly reduced the concentration of antibodies against the Botox-producing pathogen *Clostridium botulinum* in the blood of cattle indicating the suppression of the pathogen. They concluded that the neurotoxin concentration was reduced by the biochar in the gastrointestinal tract of the animals. The feeding of only 200 g of biochar per day did not show the same efficiency. However, when this lower dosage was mixed with 500 ml of lactobacilli-rich sauerkraut juice, a similar significant reduction of *Clostridium botulinum* antibodies in the blood could be measured.

*Knutson et al. (2006)* fed sheep infected with *Escherichia coli* and *Salmonella typhimurium* 77 g of activated biochar per animal per day. Although *Naka et al. (2001)* had shown earlier by in vitro trials that *E. coli* O157: H7 (EHEC) cell counts were reduced from $5.33 \times 10^6$ by five mg/ml activated biochar to below 800, the in vivo test by Knutson et al. with the same activated biochar (DARCO-KB; Norit®) revealed no biochar-related binding of either *E. coli* or *S. typhimurium* in the gastrointestinal tract of sheep. The authors hypothesized that either the biochar binding sites were occupied by competing substances or other digestive bacteria or that the time between infection with the pathogen and administration of the biochar was too long.

*Schirrmann (1984)* indicated that biochar has a particularly strong adsorption or suppression capacity for gram-negative bacteria (e.g., *E. coli*) with high metabolic activity

(see more below in section "Administration of Biochar Feed and Biochar Quality Control": Side effects of biochar). Fecal *E. coli* counts in manure after feeding 0.25% activated biochar or 0.50% coconut tree biochar were significantly lower than those of the control without biochar in 10 days finishing pig trial, while the number of beneficial bacteria *Lactobacillus* in feces increased in both biochar treatments (*Kim et al., 2017*).

Liquid cattle manure often contains *E. coli* O157: H7 (EHEC), which can contaminate water and soil and enter the human food chain (*Diez-Gonzalez et al., 1998*). Biochar can both adsorb *E. coli* and its toxic metabolites already in the digestive tract, as well as reduce the spread of those bacteria in water and soil by adding it to manure. *Gurtler et al. (2014)* investigated the effect of various biochar on the inactivation of *E. coli* O157: H7 (EHEC) when applied to soils. All biochars produced by either fast or slow pyrolysis from switchgrass, horse manure or hardwood significantly reduced EHEC concentrations, with fast pyrolysis of barley and oak log feedstock providing the best results in the contaminated soil mix, where EHEC after 4 weeks were untraceable using a cultivation based assessment (*Gurtler et al., 2014*).

*Abit et al. (2012)* investigated how *E. coli* O157: H7 and *Salmonella enterica* spread in water-saturated soil columns of fine sand or sandy loam, when the soil columns were blended with 2% of different biochars. While chicken manure biochar prepared at 350 °C did not improve the binding of either bacteria, the addition of biochar prepared at 700 °C from pinewood or from chicken manure significantly reduced the spread of both bacteria. In a later study, the authors showed significant differences in immobilization between the two bacterial strains and suggested that the surface properties of the bacteria played a significant role in the binding of these bacteria to the biochar (*Abit et al., 2014*). The latter may turn out to be an important insight into biochar—bacterial interaction and needs to be investigated systematically.

Since *E. coli* infections are likely to spread through cattle herds via water troughs, the prophylactic addition of biochar to trough water may be a preventive measure that should be further investigated.

In the study of *Watarai & Tana (2005)*, the mixture of fodder with 1% and 1.5% bamboo biochar and bamboo vinegar, respectively, slightly but significantly reduced the levels of *E. coli* and *Salmonella* in chicken excrement. A patented biochar—wood vinegar product, *Nekka-Rich* (*Besnier, 2014*), whose composition was not revealed, showed a highly significant reduction of *Salmonella* in chicken droppings. It was further found that the biochar—wood vinegar product reduced the pathogenic gram-negative *Salmonella enterica* bacteria in the droppings, but not the intestinal flora of ubiquitous, non-toxic, gram-positive *Enterococcus faecium* bacteria (*Watarai & Tana, 2005*).

A 0.3% bamboo biochar feed supplement (on DM base) suppressed the fecal excretion of gram-negative coliform bacteria and gram-negative *Salmonella* in pigs up to 20- and 1,100-fold, respectively, compared to controls without biochar (*Choi et al., 2009*). The effect of biochar on the suppression of both bacterial species was of the same order of magnitude as that of antibiotics. Feeding biochar resulted in a 190-fold increase in the number of beneficial intestinal bacteria and a 48-fold

higher level of gram-positive *Lactobacilli* compared to the treatment with antibiotics (*Choi et al., 2009*).

In vitro studies revealed that biochar, as well as clay, can efficiently immobilize cattle rotavirus and coronaviruses at rates of 79–99.99% (*Clark et al., 1998*). Since the diameter of the viral particles were larger than the pore diameters of the clay and most pores of the biochar, the authors suspected that binding was mainly due to the viral surface proteins binding to the biochar.

In vitro and in vivo experiments with bovine calves showed that biochar, especially in combination with wood vinegar, was able to control parasitic protozoan *Cryptosporidium parvum* infection and to stop diarrhea of calves within one day. The number of oocysts in the feces dropped significantly after a single day of feeding biochar; after 5 days no more oocysts could be found in the feces of the calves (*Watarai, Tana & Koiwa, 2008*). Similar results were reported when a commercial biochar wood acetic acid product (Obionekk®, Obione, Charentay, France) was tested as feed additive in young goats (*Paraud et al., 2011*). The mixture administered twice or thrice daily reduced the clinical signs of diarrhea already on the first day, and the oocyst shedding in the feces decreased significantly. Over the period of the study, the mortality of the young goats was 20% in the control group and only 6.7% in the treatment group that received Obionekk® three times per day. Biochar feeding in goats may also reduce the incidence of parasites such as cestode tapeworms and *coccidia* oocysts (*Van, Mui & Ledin, 2006*).

### Adsorption of drugs

Numerous human medical studies on the use of activated carbon in poisoning have been published in the 1980s providing important insights into the use of (activated) biochar as feed especially to treat feed poisoning (*Erb, Gairin & Leroux, 1989*). The adsorbing effect of activated carbon can be used to prevent the gastrointestinal uptake of most drugs and numerous toxins (*Neuvonen & Olkkola, 1988*), which is typically more effective than pumping out stomach contents. The repeated intake of activated carbon or biochar improved the elimination of overdosed toxicologically effective substances such as aspirin, carbamazepine, dapsone, dextropropoxyphene, cardiac glycosides and many more as summarized by *Neuvonen & Olkkola (1988)*. Moreover, a faster elimination of many industrial and environmental toxins was assessed. In acute poisoning, 50–100 g of activated biochar are administered to adults and about one g/kg of body weight to children. The same authors also point out that there are no known serious side effects from accidental ingestion. In the case of acute poisoning, Finnish physicians recommend repeated oral treatment with activated carbon to reduce the risk of toxins being desorbed from the biochar-toxin complex in the digestive cycle (*Olkkola & Neuvonen, 1989*). In general, repeated oral administration of biochar increases the efficacy of detoxication (*Crome et al., 1977*; *Dawling, Crome & Braithwaite, 1978*). However, regular administration of 0.2% activated biochar in broiler feed did not significantly impact the blood levels of the antimicrobial drugs doxycycline and tylosin, and of the coccidiostats diclazuril and salinomycin. The pharmaceutical products were co-applied to the activated carbon amended feed (*De Mil et al., 2017*).

### Adsorption of pesticides and environmental toxins

Based on the excellent adsorption properties of biochar in relation to numerous pesticides, insecticides and herbicides (*Safaei Khorram et al., 2016*; *Mandal, Singh & Purakayastha, 2017*; *Cederlund, Börjesson & Stenström, 2017*), which are increasingly found in animal feed (*Shehata et al., 2012*), biochar is considered as animal feed additive. Of particular importance is the adsorption of glyphosate, an herbicide that currently contaminates most of the feed produced from genetically modified maize, rapeseed and soybean. Although crop desiccation herbicides have been banned in Germany since May 2014, they are still permitted in many other countries as a treatment shortly before grain harvest. In addition to immobilizing magnesium and zinc, glyphosate has a potent antibiotic activity (US Patent 7,771,736, EP0001017636, issued in 2010) and is suspected of causing or promoting chronic botulism (*Shehata et al., 2012*). Glyphosate sorption efficiency onto biochar particles is both dependent on pH (high sorption at low pH; *Herath et al., 2016*) and the highest treatment temperature during biochar production (high sorption on high-temperature biochars; *Hall et al., 2018*). However, *Hall et al. (2018)* showed that glyphosate sorbed by biochar from pure water could be remobilized by adding 0.1M monopotassium phosphate solution. This finding indicates that biochar-sorbed glyphosate from feed may be remobilized in the digestive tract due to numerous ions potentially competing for sorption sites. Further research in vivo and/or in vitro in relevant matrixes is necessary, as low pH, for example, in the stomach, could favor glyphosate sorption (*Herath et al., 2016*). In a study with 380 dairy cows, *Gerlach et al. (2014)* showed that daily feeding with humic acids (120 g/day) or with a combination of 200 g of biochar and 500 g of sauerkraut juice for 4 weeks significantly reduced the glyphosate concentration in the urine of the cows that were fed with glyphosate contaminated silage.

Preliminary pesticide adsorption studies using biochar were already carried out in the 1970s (*Humphreys & Ironside, 1980*). Deposits of the systemic organophosphorus insecticide Runnel in the gastric mucosa of sheep were significantly reduced by the feeding 50 g of activated biochar per kg of feed, i.e., 5% amendment rate (*Smalley, Crookshank & Radeleff, 1971*). While it was reported that activated biochar was successfully used to adsorb pesticides in the digestive tracts of cattle, sheep and goats and were eventually excreted (*Wilson & Cook, 1970*), similar experiments in chickens did not show any significant effects on the residue levels in eggs and tissues (*Foster et al., 1972*). Feeding of biochar with Dieldrin contaminated feed, an organochloride insecticide that was widely used until the 1970s and is still persistent in the environment though it is banned now, resulted in a very significant reduction of the Dieldrin concentration in the fat of the pigs (*Dobson et al., 1971*). On the other hand, *Fries et al. (1970)* found no reduction in the levels of Dieldrin and DDT in milkfat when cows were fed one kg of activated biochar per day for 14 days. However, *Wilson et al. (1971)* found that when Dieldrin and DDT-contaminated feed was mixed with activated biochar at 900 g per animal and day, Dieldrin intake was reduced by 43% and DDT intake by 24%. When the contaminated feed and biochar were administered separately, DDT intake was not reduced as both the Dieldrin and DDT were probably absorbed by the oral mucosa already and not only in

the digestive tract (*Fries et al., 1970*). Activated biochar also showed very good in vitro adsorption properties for the herbicide Paraquat (*Okonek et al., 1982*; *Gaudreault, Friedman & Lovejoy, 1985*), which has been banned in the EU since 2007 but is still legal in the US and other countries.

Fat-soluble organochlorine compounds such as Dibenzo-*p*-dioxin (PCDDs), Dibenzofuran (PCDFs) and dioxin-like PCBs are ubiquitous environmental toxins, and can often be detected in animal feed. These compounds accumulate in the adipose (fatty) tissue of animals and humans. Experiments with activated biochar to adsorb these substances were undertaken repeatedly in Japan (*Yoshimura et al., 1986*; *Takenaka, Morita & Takahashi, 1991*; *Takekoshi et al., 2005*; *Kamimura et al., 2009*). All experiments showed the strong affinity of the organochlorine compounds to activated biochar (*Iwakiri, Asano & Honda, 2007*). *Fujita et al. (2012)* carried out an extensive experiment with 24 laying hens whose feed contained the organochlorine compounds mentioned above and fed either with or without 0.5% biochar over a period of 30 weeks. Depending on the structure and aromaticity of the organochlorine compounds, concentrations of PCDDs/PCDFs, non-ortho PCBs and mono-ortho PCBs in the tissue and eggs of the laying hens could be reduced by more than 90%, 80% and 50%, respectively (*Fujita et al., 2012*). The fact that different organochlorine compounds are bound to different degrees by biochar has been previously demonstrated in studies of contaminated fish oil (*Kawashima et al., 2009*). In general, molecules with higher aromaticity have a stronger affinity to biochar; this also applies to polycyclic aromatic hydrocarbons (*Bucheli, Hilber & Schmidt, 2015*). *Olkkola & Neuvonen (1989)* concluded that the regular intake of biochar as food supplement can be very helpful in the elimination of industrial and environmental toxins including dioxins and PCB ingested by humans, a valid statement for animal feed too.

### Detoxification of plant toxins

Another benefit of a regular use of biochar is the alleviation of adverse effects of naturally occurring though potentially harmful ingredients such as tannins contained in many feeds (*Struhsaker, Cooney & Siex, 1997*). Tannins are complex and extraordinarily diverse compounds that are partly beneficial but may also be harmful especially to ruminants. Tannins are often found in high protein feeds such as legumes and the strong taste repels the animals, which reduces digestability and weight gain (*Naumann et al., 2013*). Several studies have investigated how biochar feeding alters the impact of tannin-rich foods. *Van, Mui & Ledin (2006)* found that in goats, feeding 50–100 g of bamboo biochar per kg of a tannin-rich acacia leaf diet increased daily weight gain by 17% compared to the control without biochar. The authors found that digestion of crude proteins and nitrogen conversion were significantly improved. Apparently, there was an optimal biochar dose: While 50 and 100 g of bamboo biochar feed additions resulted in similar goat weight gains, feeding 150 g of the same biochar per kg diet did not show any improvement compared to control. *Struhsaker, Cooney & Siex (1997)* found, as previously described, that the consumption of wild fire derived charcoal by Zanzibar red colobus monkeys increased the nutritional efficiency of tannin-rich Indian almond and mango leaves. *Banner et al. (2000)*

found that the mixture of 10–25 g of activated biochar per day with rye significantly increased the uptake of tannin and terpene rich compounds. Similar results for sage and other terpenic and tannin-rich shrubs were reported by *Rogosic et al. (2006, 2009)*, whereas others could not confirm that lambs consumed significantly more sage due to biochar amended feed (*Villalba, Provenza & Banner, 2002*).

In winter, when hardly any fresh pasture plants are available, sheep also eat bitterweed (*Hymenoxys odorata* DC.), which contains toxic levels of sesquiterpene lactones. *Poage et al. (2006)* conducted therefore a series of bitterweed feeding trials with 0.5–1.5 g of biochar per lamb per day mixed directly to the feed. While the lambs rejected the bitterweed-containing feed without biochar, they did consume bitterweed up to 26.4% of the total feed intake when combined with biochar revealing no signs of toxicosis.

Several studies have shown that poisoning of both livestock and sheep through contamination of feed with *Lantana camara, a species of flowering invasive species*, can be effectively treated with five g of biochar per kg of body weight (*Pass & Stewart, 1984*; *McLennan & Amos, 1989*). While five out of six calves recovered from *Lantana camara* poisoning after treatment with activated biochar, five out of six calves not treated with biochar died (*McKenzie, 1991*). Treatment with bentonite achieved similarly high cure rates, but complete healing took about twice as long. Similarly, significant results are found for treating Yellow tulip (*Moraea pallida*) poisoning of cattle (*Snyman et al., 2009*) and oleander poisoning of sheep (*Tiwary, Poppenga & Puschner, 2009*; *Ozmaie, 2011*).

## Regular biochar feeding to improve performance and animal welfare

While therapeutic administration of biochar is a historically proven practice and has been scientifically studied for over 50 years and recommended as a cure for numerous symptoms, regular co-feeding of biochar with the purpose of improving productivity is discussed again only since 2010. The feeding of livestock with biochar and biochar products is rapidly spreading in practice, due to the apparently good experiences of farmers, especially in Germany, Switzerland, Austria and Australia. However, systematic scientific research on regular feeding with various types of biochar is still rare. One reason for this is the fact that with veterinary medicine and biochar research two areas of expertise collide that could hardly be more different and whose methods and vocabulary have little in common. The latter also explains why usually non-characterized or only poorly characterized biochar was used for feeding experiments.

Despite the diversity of biochar properties, key features of this heterogeneous material are similar and apparently lead to comparable effects when provided as feed supplement. The review of 27 peer reviewed scientific publications and clinical studies (Table 1) about regular biochar feeding revealed no negative effects on animal welfare and performance. Still, there are open question on some effects on long-term biochar feeding that should be addressed prior to an unconfined recommendation of regular biochar feeding. These include effects on the resorption of liposoluble feed ingredients and potential interaction with the mycotoxin fumonisin. These risks of regular biochar feeding are summarized in a separate section below. While results of feeding trials were sometimes neutral (no significant difference between biochar and control treatment), often one or

**Table 1 Overview of published studies on biochar feeding.**

| Animal | Daily BC intake | Feedstock | HTT in °C | Activation | Blend | Weight increase in % | Duration in days | Other results and remarks | Source |
|---|---|---|---|---|---|---|---|---|---|
| Cattle | 0.6% of feed DM | Rice hull | 700 | No | | 25 | 98 | Reduced enteric methane emissions | Leng, Inthapanya & Preston (2013) |
| Bull | 2% of feed DM | Wood | >600 | No | Vitamin A | n.s. | | | Kim & Kim (2005) |
| Cattle | 1% of feed DM | Rice husk | >600 | No | | 15 | 56 | 15% feed conversion rate increase | Phongphanith & Preston (2018) |
| Goat | 1% of body weight | Bamboo | | No | | 20 | 84 | DM, OM, CP digestibility and N retention increased | Van, Mui & Ledin (2006) |
| Goat | 1% of feed DM | | | No | | 27 | 90 | DM, OM, CP digestibility and N retention increased | Silivong & Preston (2016) |
| Pig | 0.3% of feed DM | Bamboo | >600 | Yes (900) | Bamboo vinegar | 17.5 | 42 | Improved the quality of marketable meat | Chu et al. (2013c) |
| Pig | 0.3% of feed DM | Wood | | No | Stevia | 11 | | Higher meat quality and storage capacity | Choi et al. (2012) |
| Pig | 1%, 3% and 5% of feed DM | Wood | 450 °C | No | 25% wood vinegar | n.s. | 30 | Increased duodenal villus height | Mekbungwan, Yamauchi & Sakaida (2004) |
| Pig | 1% of DM feed | Wood | >600 | No | Lactofermented | n.s. | 28 | | Kupper et al. (2015) |
| Pig | 1% of DM feed | | >500 | No | | 20.1 | 90 | 20.6% increased feed conversion rate | Sivilai et al. (2018) |
| Poultry | 0.2% of DM feed | Wood | | No | | 17 | 49 | | Kana et al. (2010) |
| Poultry | 0.2% of DM feed | Maize cob | | No | | 6 | 49 | Improved carcass traits | Kana et al. (2010) |
| Poultry | 2%, 4%, 8% of feed DM | Citrus wood | | No | | 0 | 42 | Heavier abdomen fat | Bakr (2007) |
| Poultry | 2.5%, 5%, 10% of feed DM | Wood | | No | | 0 | 42 | Weight increase up to 28 days but not after 49 days | Kutlu, Ünsal & Görgülü (2001) |
| Poultry | 0.3% of feed DM | Wood | | No | | 3.9 | 140 | Reduced mortality by 4% | Majewska, Pyrek & Faruga (2002), Majewska, Mikulski & Siwik (2009) |
| Duck | 1% of DM feed | Bamboo | >650 | No | Bamboo vinegar | n.s. | 49 | Intestinal villus height increased | Ruttanavut et al. (2009) |

(Continued)

| Animal | Daily BC intake | Feedstock | HTT in °C | Activation | Blend | Weight increase in % | Duration in days | Other results and remarks | Source |
|---|---|---|---|---|---|---|---|---|---|
| Duck | 1% of DM feed | Wood | | No | Kelp | n.s. | 21 | Feed conversion rate increased | *Islam et al. (2014)* |
| Poultry | 4% of DM feed | Woody green waste | 550 | No | | n.s. | 161 | Egg weight increased by 5%; feed conversion ratio by 12% | *Prasai et al. (2016)* |
| Poultry | 1% of DM feed | Rice husk | >550 | No | | n.s. | | Reduced pathogenes in feces | *Hien et al. (2018)* |
| Poultry | 0.7% of DM feed | Wood | >650 | No | Lactofermented | n.s. | 36 | | *Kupper et al. (2015)* |
| Poultry | 1% of DM feed | Wood | >650 | No | Lactofermented | 5 | 37 | Reduced foot pat and hook lesions by 92% and 74% | *Albiker & Zweifel (2019)* |
| Flounder | 0.5% of DM feed | Bamboo | | No | | 18 | 50 | Feed and protein conversion rate increased | *Thu et al. (2010)* |
| Flounder | 1.5% of DM feed | Wood | | No | 20% wood vinegar | 11 | 56 | Highest feed efficiency increase of 10% at 0.5% BC | *Yoo, Ji & Jeong (2007)* |
| Stripfish | 1% of DM feed | Rice husk | >600 | No | | 36 | 90 | Significantly improved water quality | *Lan, Preston & Leng (2018)* |
| Stripfish | 1% of DM feed | Wood | | No | | 44 | 90 | Significantly improved water quality | *Lan, Preston & Leng (2018)* |
| Carp | 0.5%, 1%, 2%, 4% of DM feed | Bamboo | | No | | n.s. | 63 | Improved serum indicators | *Mabe et al. (2018)* |
| Stripfish | 2% of feed DM | Bamboo | | No | High VOC biochar | 27 | 50 | Survival rate increase by 9% | *Quaiyum et al. (2014)* |
| | | | | | Mean | 9.9 | | | |

**Note:**
The table indicates the percentage weight increase of various livestock depending on the ingested biochar type and daily feed intake. A total of 61% of the 28 data set delivered weight increases while the remaining trials did not result in significant increases.

several of the following effects were observed when biochar was provided as feeding additive to livestock:

- Increase in feed intake
- Weight gain
- Increased feed efficiency
- Higher egg production and quality in poultry
- Strengthening of the immune system
- Improvement of meat quality
- Improvement of stable hygiene and odor pollution
- Reduction of claw and feet diseases
- Reduction of veterinary costs

Sorted by animal species, the following subsection reviews the scientific literature on medium to long term feeding of biochar in regard to improving livestock productivity, product quality, animal fitness, welfare and performance in the respective animal farming system. Risks of regular biochar feeding are summarized in a separate section.

### Cattle

As evidenced by farmer practice, veterinary advice, and European regulations, biochar is already widely used as a regular feed supplement in cattle farming especially in Germany, Austria and Switzerland (European Biochar Certification body, Hans-Peter Schmidt, 2018, personal communication). However, there are only very few scientific studies on biochar feed additives for cattle so far.

Since 2011, the German veterinarian Achim Gerlach has been feeding 100–400 g of high temperature wood biochar (HTT 700 °C) per cow per day to numerous herds of cattle without detecting negative side effects (*Gerlach & Schmidt, 2012*; Hans-Peter Schmidt, 2018, personal communication). His survey of 21 farmers with at least 150 cattle revealed that overall health and vitality had improved since they had started biochar feeding. The somatic cell count (SCC) of the milk, an indicator of level of harmful bacteria, decreased significantly, whereas milk protein and milk fat content increased. When biochar additions to feed stopped, SCC quickly increased and a general loss of performance of the animals compared to the biochar-feeding period was observed. It was also reported that hoof problems were reduced, and that postpartum health was stabilized through biochar co-feeding. Within 1–2 days after the onset of the biochar feeding, diarrhea symptoms decreased and feces became firmer. Mortality rates declined, as did overall veterinary costs. The liquid manure viscosity improved significantly and the odor load of the manure decreased (*Gerlach & Schmidt, 2012*).

For 98 days, Leng, Preston & Inthapanya fed four cattle 0.6% of a rice hull-derived biochar, with another four in a control group without biochar in their feed. The biochar feeding resulted in a 25% higher weight gain compared to the control animals (*Leng, Preston & Inthapanya, 2013*). Another study, however, did not find any significant effect on weight gain and blood values in Hanwoo bulls when an undefined biochar

was administered at a rather high dose of 2% (*Kim & Kim, 2005*). A supplement of 1% rice husk biochar was added to a basal diet consisting of ensiled cassava root, urea, rice straw and fresh cassava foliage (*Phongphanith & Preston, 2018*). Live weight gain increased by 15% and feed conversion rate also improved by 15% in the biochar treatment, compared to the control without biochar supplement. Interestingly, when a rice wine distillers' byproduct was added at 4% to the biochar-supplemented feed, the live weight gain and the feed conversion rate increased by 60% compared to the control without either supplement. They further found an increase of 18% compared to feeding with the rice wine distillers alone (without biochar), or 31% compared to the biochar-only supplement. This shows a strong interactive effect between the two supplements indicating that the combination and interaction of biochar with other feed additives should increasingly be investigated.

In a semi-continuous artificial rumen system, a high temperature biochar (HTT 600 °C) was added at 0%, 0.5%, 1% and 2% to a high-forage diet for 17 days. The biochar linearly increased the digestion of dry matter, organic matter, crude protein and fiber. Microbial protein synthesis also increased linearly. The microbial production of acetate, propionate and total volatile fatty acids in the artificial rumen increased (*Saleem et al., 2018*).

As early as 2010, Marc McHenry pointed to the possibility of using biochar as a feed additive not only to increase feed efficiency but to also increase nutrient availability of the manure, to protect ground and surface water, and to sequester carbon in the soil (*McHenry, 2010*). This cascading approach of improving not only animal performance and welfare but also various ecosystem services has been the subject of discussion and investigation by various authors since (*O'Toole et al., 2016*; *Schmidt & Shackley, 2016*; *Kammann et al., 2017*). A far-reaching study of these cascades has been carried out by *Joseph et al. (2015b)* in Australia: Since 2011, 60 grazing cattle on an Australian farm were fed 330 g per day of a high temperature biochar (HTT 600 °C) made from Jarrah wood mixed with 100 g of molasses. From 2011 to 2015, soil organic matter, pH (CaCl$_2$), Colwell-P, Colwell-K, electrical conductivity and the content of all exchangeable cations increased in the pasture soil that received the dung of the free ranging cattle. During its passage through the digestion system of the cattle, biochar seems to capture organic and mineral compounds with high plant fertilizing properties that would otherwise probably be subject to rather quick leaching during storage. Most of these captured plant nutrients (especially nitrogen and phosphorus) remain bound in the porous structure of the biochar until its incorporation into the soil, where they likely become, to a large extent, plant available as has also been found for biochar after aerobic composting (*Kammann et al., 2015*; *Schmidt et al., 2017*). The authors of the Australian study reported that increased retention of the digested nutrients in the biochar increased the fertilizing effect of the bovine manure so that no additional fertilizers was required for the pasture growth (*Joseph et al., 2015b*). However, they did not set-up a control pasture to proof the latter. To prove their conclusion, a more systematic scientific experiment would be required.

In addition to the improvement of the fertilizing properties of biochar-amended manure, the application of biochar to manure either via feed or via bedding materials is

recommended as a potent strategy to reduce manure related greenhouse gas emissions (*Kammann et al., 2017*). When biochar (wood shavings, HTT 650 °C) was applied at 13% to a cattle slurry and subsequently applied to a field at 3.96 $m^3$ biochar $ha^{-1}$, the biochar decreased total $NH_3$-emissions by 77%, $N_2O$-emissions by 63% and $CH_4$-emissions by 100% compared to the control of cattle slurry only (*Brennan et al., 2015*).

Since 2012, German and Swiss farmers have been using biochar in the production of feed silage to stabilize lactic acid fermentation, prevent fermentation failure and reduce risks of fungal infestation and formation of mycotoxins (*O'Toole et al., 2016*). Lower levels of acetic acid and especially butyric acid are expected to minimize the risk of *Clostridia* infestation. The high-water holding capacity of biochar appears to buffer the water content of the silage, reducing the formation of excess fermentation liquids.

*Calvelo Pereira et al. (2014)* investigated the addition of various amounts and types of biochar (0–2.1–4.2–8.1–18.6% made from pine wood or maize straw and pyrolyzed at 350, and 550 °C, respectively) to hay silage and to cattle rumen liquid. The biochar treatments did not significantly affect the investigated silage quality parameters, nor did it negatively affect in vitro incubation with rumen fluid.

### Goats and sheep

In a 12-week experiment with 42 young goats, it was found that feeding one g of bamboo biochar per kg of bodyweight resulted in significantly higher crude protein intake (*Van, Mui & Ledin, 2006*). The total amount of digested nitrogen increased and was thus lower in the urine and feces of the animals. The body weight increased on average 53 g per day compared to 44 g in the control group fed without biochar; a statistically significant difference of 20%. The basic feeding of the goats included a large proportion of tannin-rich acacia (*Acacia mangium*) leaves, and the authors hypothesized that biochar eased digestion of those leaves by sorption of their tannins which apparently lead to higher crude protein and improve total DM intake.

In a trial with groups of 12 goats ($N$ = 3), growth performance was tested when a basal diet of tannin rich leaves of *Bauhinia acuminata* were provided either with or without 1% biochar (*Silivong & Preston, 2016*). Biochar improved the nutrient assimilation and led to a 27% increase in daily weight gain over the 100-day period of the trial. In another study, a goat feed additive of 1.5% and 3% activated coconut biochar did not produce significant improvement of feed intake nor did it alter the microbial community structure compared with the control (*Al-Kindi et al., 2017*). However, the activated biochar increased the fecal concentration of slowly decomposable carbohydrates while reducing fecal N. This left the authors to surmise a beneficial slow-down in the mineralization rate of the organic carbon contained in the manure when applied to soil, which may be beneficial for the built-up of soil organic matter.

### Horses

Very few publications exist yet on feeding biochar to horses. *Edmunds et al. (2016)* investigated the effect of a woody biochar on the microbial community of the equine hindgut and the metabolites they produce. They did not find any significant effect of the

biochar and concluded that the effect of biochar as a control for toxic substances is at its highest in the foregut or midgut of animals, and therefore should have little impact on the hindgut.

According to the EBC certified manufacturers of biochar and biochar products, horse breeders and farmers widely apply biochar in horse manure management and also in feeding, but apart from the above, not a single scientific study is known to the authors.

### Pigs

Chu et al. published several fundamental studies in 2013 on the feeding of bamboo biochar to pigs. Young pigs ($N = 12$) were fed for 42 days in addition to their normal fattening diet (corn, wheat, soybean meal) either with 0%, 0.3% or 0.6% of biochar. The average weight gain during the trial period was 750 g per day in the control without biochar and 877 g per day in the 0.3% biochar treatment; this corresponded to a significant feed efficiency increase of 17.5%. Doubling the biochar supplement to 0.6% did not lead to statistically significant differences compared to the 0.3% treatment. While leucocytes, erythrocytes, hemoglobin, hematocrit and platelets did not differ significantly between the experimental groups, the biochar group showed significant positive effects on total protein, albumin, cholesterol, HDL-CH and LDL-cholesterol levels in the blood plasma. In addition, the cortisol content was significantly lower, which indicates a reduced susceptibility to stress (*Chu et al., 2013c*). In another study, the authors showed that feeding 0.3% and 0.6% bamboo biochar improved the quality of marketable meat and the composition of pig fat, with an increase in unsaturated fatty acid content and a decrease in saturated fat (*Chu et al., 2013b*). In a third study, the authors examined to what extent biochar feeding can replace the regular supplementation of growth-promoting antibiotics, something which is still legal in many though not all countries. In a very comprehensive publication (*Chu et al., 2013a*), they concluded that feeding 0.3% bamboo biochar gave the same growth rate in fattening pigs as the standard antibiotic treatment, notably without the negative side effects to the environment that antibiotics can have.

Another hog feed trial was done in South Korea using different concentrations of biochar and stevia mixed into the common diet of 420 pigs (*Choi et al., 2012*). While neither 30 g of biochar nor 30 g of stevia per kg of feed alone had any significant effects, 30 g of biochar plus 30 g of stevia had higher daily weight gain, feed efficiency and immune responses as well as significantly higher meat quality and storage capacity of meat products (*Lee et al., 2011*; *Choi et al., 2012*). In a Japanese study by *Mekbungwan, Yamauchi & Sakaida (2004)*, piglets were fed with increasing concentrations of a 4:1 mixture of a low temperature biochar (HTT 450°) and wood vinegar. When fed with 1%, 3% and 5% of this mixture, no statistically significant effects on body weight and feed efficiency were observed compared to the 0% control. However, duodenal villi height, an animal health indicator, increased significantly. The same authors showed 4 years later, with the same biochar-wood vinegar mix added at 1% and 3% to a protein-rich feed, that the biochar treatments prevented negative side-effects of pig fattening with protein-rich pigeon peas (*Mekbungwan et al., 2008*). The biochar-fed animals presented

significantly better values in parameters related to health such as intestinal villi height, cell area and cell mitosis number compared to the control groups.

In Switzerland, *Kupper et al. (2015)* fed 80 weaned piglets for 28 days with a 1% commercial biochar feed additive mixture that had undergone a lactic fermentation beforehand. The biochar treatment did not reveal any significant difference in daily weight gain, feed consumption and feed conversion rate compared to the control group that received the same feed but without the biochar containing supplement. Moreover, no significant difference in $NH_3$-emissions of the stored or field applied manure was observed.

In a trial with native Moo Lath pigs ($N = 20$), the addition of 1% biochar to a basal diet consisting of ensiled banana pseudo stem and ensiled taro foliage increased the feed conversion rate by 10.6% compared to the control. The total weight gain of the piglets was on average higher by 20.1% ($p = 0.089$) after the 90 days of the experiment (*Sivilai et al., 2018*).

### Poultry

Of all publications on the performance-enhancing use of biochar, a majority have focused on its use with poultry, not least because scientific studies using poultry are easier and less costly to perform than on large ruminants or pigs. One of the more frequently cited studies is that of *Kana et al. (2010)* who systematically fed two different biochars, one from corncobs and the other from canary tree (*Bakeridesia integerrima*) seeds, to broiler chickens at different feeding concentrations from 0% to 1% per kg feed. Unfortunately, the production of biochar was only designated as "traditional" and was not described in detail, but the high ash levels of 47% and 25%, respectively, indicate that a substantial portion of the initial biomass was burned and not fully pyrolyzed. Nevertheless, feeding both biochars up to 0.6% led to greater, mostly significant weight gain, while the higher dosages led to no further significant weight gain, but also to no weight loss compared to the control. Liver weight, abdominal fat nor bowel length and weight were affected by the biochar feeding. The study is an important indication that biochar derived from non-woody biomass and with a higher ash content may also be suitable for feeding, which is so far not allowed by the *European Biochar Foundation (EBC) (2012)*. In a later study with the same biochars, the authors examined whether chickens can, thanks to the biochar supplement, be fed with 20% chickpeas, a feed that is protein-rich but generally difficult for chickens to digest. Surprisingly, when the ash-rich biochar from corncobs was added, the boiled chickpeas could be fed and provided the same weight gain in the broilers as the control without chickpeas. However, the lower-ash biochar from the tree seeds did not show the same effect here (*Kana, Teguia & Fomekong, 2012*).

*Bakr (2007)* used traditionally produced citrus wood charcoal purchased at the local market in Nablus and added them at very high dosages of 0%, 2%, 4% and 8% to the standard broiler feed. At 2%, significant increases on body weight, feed intake and feed efficiency were measured during the first three weeks compared to control. After this initial period, all results were similar. Of particular note in this study is that even the very high feeding dosage of 8% of a biochar of at least doubtful quality did not cause any adverse effects. *Kutlu, Ünsal & Görgülü (2001)* also used very high biochar dosages of up to 10%

of the base diet, and found that all dosages significantly increased basal feed intake in the first 28 days, and also weight gain and feed efficiency of both broilers and laying hens but did not show significantly higher gains after this initial period.

A Polish working group led by Teresa Majewska conducted several feed trials on chickens and turkeys between 2000 and 2012 (*Majewska & Pudyszak, 2011*; *Majewska, Mikulski & Siwik, 2009*; *Majewska, Pyrek & Faruga, 2002*). They achieved consistently positive results with doses of 0.3% of a hardwood biochar. They not only found higher weight gain and better feed efficiency, but also higher protein levels in the pectoral muscles and a significantly lower mortality compared to the control. Majewska et al. explained these improvements by (1) the detoxification of feed components, (2) the reduction in surface tension of the digestive pulp and (3) the improvement in fat loss in the liver.

*Ruttanavut et al. (2009)* did not find a statistically significant increase in duck growth when co-fed with a 1% biochar—wood vinegar blend, but they showed significant biochar effects on the size of the villi, the cell surface, and the rate of cell division in the gut, which confirms similar results from literature (*Samanya & Yamauchi, 2001*; *Ruttanawut, 2014*). *Islam et al. (2014)* showed in an experiment with 150 young ducks that feeding with 1% of a 1:1 mixture of biochar and sea tangle (*Laminaria japonica)* can be recommended as an alternative to the use of antibiotics in the feeding of ducks.

Several research groups have shown that the quality of chickens' meat can be significantly improved by feeding of biochar (*Cai, Jiang & He, 2011*; *Kim et al., 2011*; *Yamauchi, Ruttanavut & Takenoyama, 2010*; *Yamauchi et al., 2014*). It was for example found that no significant weight gain was recorded when fed with 0.5% activated coconut shell biochar but that Serum Glutamine, Oxaloacetic Transminase, Serum Glutamine Phosphate Transminase, Albumin and triglycerides as well as sensory evaluation and weight of abdominal fat, heart and spleen significantly improved while the cholesterol level decreased (*Jiya et al., 2013*, *2014*). Also, when broiler chickens were fed with 1% activated biochar the useful fatty acid, oleic acid and total mineral content of the meat increased significantly (*Park & Kim, 2001*). Other trials with 2% biochar or a mixture of bamboo biochar and wood vinegar did not show significant differences in meat quality compared to controls (*Sung et al., 2006*; *Fanchiotti et al., 2010*; *Ruttanawut, 2014*).

It was observed in several studies that the strength of eggshells can be improved by co-feeding biochar (*Kutlu, Ünsal & Görgülü, 2001*; *Ayanwale, Lanko & Kudu, 2006*; *Kim et al., 2006*). *Yamauchi, Ruttanavut & Takenoyama (2010)* found an increase in egg production of nearly 5% when hens were fed with a blend of bamboo biochar and wood vinegar. The collagen content of the eggs increased highly significantly by 33% with a 1% feed of the same bamboo biochar—wood vinegar mixture. Collagen not only increases the shelf life of the eggs but is also an interesting ingredient for pharmaceuticals and cosmetics (*Yamauchi et al., 2013*).

*Prasai et al. (2016)* investigated biochar, bentonite and zeolite for selective pathogen control in hens. Their treatments involved the commercial layer diet (control group) amended with biochar, bentonite and zeolite at 4% w/w, respectively. While bird weight and number of eggs did not differ significantly between the control and the biochar treatment, the total egg weight increased by 5% and the feed conversion ratio increased by

12% compared to the control. Feeding bentonite and zeolite revealed comparable increases and non-significant differences to biochar, respectively. The biochar feed amendment did not result in altered gut microbial community richness and diversity compared to the control. However, individual phylotypes at different phylogenetic levels did respond differently to the three amendments and reduced especially the abundance of *Helicobacter* and *Campylobacter*. Both genera are gram-negative and include multiple pathogenic species. The authors demonstrated that biochar, bentonite and zeolite can be used to selectively reduce the abundance of some major poultry zoonotic pathogens without reducing chicken microbiota diversity or causing major shifts in the gut microbial community and are thus a viable alternative to antibiotics in the poultry industry. A recent Vietnamese study on supplementing chicken feed with 1% rice husk biochar confirmed positive effects on pathogen occurrence with reduced plasma triglycerides, total coliform bacteria in litter and *E. coli* in feces (*Hien et al., 2018*). However, no impact on live weight gain, feed consumption and feed conversion ratio were observed.

In Switzerland, two groups of 400 broilers were fed for 36 days with a 0.7% biochar supplement provided as a commercial feed additive mixture that had undergone a lactic fermentation beforehand (*Kupper et al., 2015*). The biochar treatment did not reveal any significant difference in daily weight gain, feed consumption, feed conversion rate or food pat and hook lesions compared to the two control groups that received the same feed without the biochar containing supplement. Moreover, no significant difference in $NH_3$-emissions of the stored or field applied broiler manure was measured. The results of *Kupper et al. (2015)* are in puzzling contradiction with a similar trial in the same country undertaken at the Swiss Aviforum where groups of 270 broilers with four replicates were fed for 37 days with the same 0.9% biochar based commercial feed additive, with 1% pure wood based biochar (HTT of 700 °C) or with 0% biochar as control group (*Albiker & Zweifel, 2019*). Here, the weight gain increased significantly by 5% (fermented biochar product) and 6% (pure biochar) compared to the control. Moreover, both biochar treatments decreased the foot pat and hook lesions by 92% and 74%, respectively, compared to the control.

For a study at West Virginia University with test groups of 1,472 broiler chicks ($N = 8$), pyrolyzed poultry manure was provided as feed additive despite insufficient feed quality analyses (*Evans, Boney & Moritz, 2016*). The arsenic content of the poultry manure biochar exceeded the threshold of the European Biochar Feed Certificate (*European Biochar Foundation (EBC), 2018*) by a factor of 6.5, and no PAH analyses were carried out, despite using gasification technology that is known for the risk of producing biochars with high levels of PAH contaminations which often exceed threshold values of the EBC by factor 100 and more (*Hilber et al., 2012*; *Bucheli, Hilber & Schmidt, 2015*). Irrespective of these issues, supplementing poultry manure biochar at 2% increased the feed conversion ratio by 7% while at 4% biochar supplementation the life weight gain decreased by 8% both compared to the control. No other investigated parameter showed significant differences to the control over the 21-day experimental period. The feeding of such pyrolyzed material is in several regards not in agreement with the EBC-feed standard, and feeding uncharacterized excrement-based materials is certainly not up to ethical standards.

In an Australian trial, groups of 20 layer hens ($N = 4$) were fed a biochar made at 550 °C from green wood waste at rates of 0%, 1%, 2% and 4%, respectively (*Prasai et al., 2018a*) for 25 weeks. While no significant difference in weight gain was observed, the feed conversion ratio improved significantly between 10% and 13% in the three biochar treatments compared to the control without biochar. The egg weight was 5% higher in the 2% biochar treatment and 4% higher in the 4% treatment compared to the control. Standardized indicators of egg quality (i.e., Haugh unit, Albumen height, stability of egg shell) where not changed by the biochar feed amendment. The Yolk color index, however, decreased with increasing biochar dosage. The same effect was also found when bentonite or zeolite was used instead of biochar. Yolk color is mainly the result of carotenoid content (*Bovšková, Míková & Panovská, 2014*). Carotenoids are lipophilic organic molecules that accumulated from the feed. Thus, we hypothesize that biochar may sorb a certain amount of lipophilic ingredients of the feed. The N-balance between feed-N intake, egg-N, excreta-N and lost N did not differ significantly between the treatments though the excreta-N was reduced by 20–34% in the 2% and 4% biochar treatment compared to the control. The lower recovery of N in excreta is indicative of a more efficient digestive extraction of N, consistent with the observed higher feed conversion efficiency. Remarkably, the inclusion of 2% and 4% biochar maintained egg production at normal levels when birds were challenged with fungal-contaminated feed. In the control treatment, the contaminated feed led to decreased egg production by 16%. The same main author found, in another publication based on a similar trial with the same 1%, 2% and 4% biochar amendments, improvements of the poultry manure especially in regard to granule size, water retention and decomposition characteristics (*Prasai et al., 2018b*). N-contents in the decomposed poultry manure were lower by 20% and 26%, respectively, in the treatment with 2% and 4% biochar feed compared to the control. $NH_3$-emissions of the manure, measured in a separate experiment using incubated bell jars, increased by 31% in the treatments with 2% and 4% but not with 1% biochar feed amendments compared to the control. This increase in ammonia emissions due to high doses of poultry feed applied biochar is puzzling as the addition of higher dosages (5–15% (m/m)) of biochar to poultry manure composting was shown to decrease ammonia emissions between 53% and 89% (*Rong et al., 2019*). Apparently, biochar affects poultry manure composting differently when applied to the feed versus when applied directly to the manure.

### Aquaculture

Nowadays aquaculture provides as much product for human consumption as capture fisheries, yet it causes considerable harm to the environment if effluents with fish feces and excess feed nutrients are not treated and recycled into valuable fertilizers (*UN, 2016*). Biochar supplements have been fed to fish with the intention to improve water quality as well as fish health and productivity. Japanese flounder were fed with 0–4% incremental doses of a bamboo biochar mixed into the regular feed (*Thu et al., 2010*). While all biochar feed additions resulted in significantly higher flounder weight gains, the variability of individual results was so high that only the 0.5% dose provided statistically significantly

higher weight gain rates of 18%. It was noteworthy that all biochar feeding rates resulted in significantly lower nitrogen excretions and reduced the nitrate content in the fish water by >50%. In a South Korean experiment also with flounder, dosages from 0% to 2% of a biochar—wood vinegar blend were fed. At a dose of 1%, the feed efficiency increased significantly by 10%, and also the total weight gain of the fish was significantly higher (*Yoo, Ji & Jeong, 2007*). The authors concluded that feeding rates between 0.5% and 1% of DM feed intake may deliver maximum weight gain and feed efficiency.

Two different biochars, one made from rice husks in a TLUD stove (*Anderson, Reed & Wever, 2007*) and one made from wood in traditional charcoal kilns, were compared as a 1% feed additive for tank raised striped catfish (*Pangasius hypophthalmus*) (*Lan, Preston & Leng, 2018*). Growth rates increased by 36% with the rice husk biochar and 44% with the wood biochar compared to the control. Both biochars led to 25% increased ratio of weight to length indicating an enhanced flesh to bone ratio due to the faster growth rate caused by the biochar additive. Water quality improved significantly as levels of ammonia nitrogen, nitrite, phosphate and chemical oxygen demand decreased by 24%, 22%, 15%, 21%, respectively, in the rice husk biochar treatment with similar values for the other biochar. The authors hypothesized that biochar may facilitate the formation of biofilms as habitat for gut microbiota which could be the explanation for the improved growth rates.

In China, a dietary bamboo biochar was added to the feed of juvenile common carps at rates from 1% to 4% (*Mabe et al., 2018*). The biochar treatments did not produce any obvious effect on the growth performance of the carps compared to 0% control. However, significant improvements were reported on serum indicators such as alanine aminotransferase, aspartate aminotransferase, total protein, triglycerides, total cholesterol, high density lipoprotein (HDL) and glucose, demonstrating an increase in fish quality and health. The most beneficial effects were found at the highest biochar dosage. No adverse effects were observed.

## Reduction of methane emissions from ruminants

Ruminant production accounts for about 81% of the total GHG from the livestock sector (*Hristov et al., 2013*). While in chickens, pigs, fish and other omnivores most of the greenhouse gas emissions are caused by the decomposition of solid and liquid excretions, ruminants' GHG emissions are mainly produced by direct gaseous excretions through flatulence and burping (eructation). The latter mainly affects cattle which are capable of producing 200–500 l of methane per day (*Johnson & Johnson, 1995*). These methane emissions, mainly produced through rumen microbial methanogenesis, are responsible for 90% of the GHG caused by cattle (*Tapio et al., 2017*).

In the bovine rumen, methanogenesis is carried out by archaea that convert microbial digestion products $H_2$ and $CO_2$ or formate (HCOOH, methanoate) to $CH_4$ to gain energy under anoxic conditions. While hydrogen serves as an electron donor for the microbial reduction of $CO_2$ to methane ($CH_4$), the reduction of formate (requiring six electrons to be reduced to $H_2$ and $CO_2$) can have several biochemical pathways. The production of methane means a significant loss of energy for the animal (from 2% to 12% of the total

energy intake; *Tapio et al., 2017*) as the high-energy methane cannot be digested any further and has to be eliminated almost entirely through eructation (burp) and only minimally via flatulence from the digestive tract (*Murray, Bryant & Leng, 1976*). Since methane is a 28–34 times more harmful than $CO_2$ (global warming potential with and without climate-carbon feedbacks over a period of 100 years; *Myrhe et al., 2013*), there is an increasing interest in feed supplements that not only increase feed efficiency, but also can reduce methane emissions resulting from ruminant digestion.

Numerous studies have sought to find other electron acceptors besides $CO_2$ and enteric fatty acids to reduce methanogenesis. However, until recently, apart from the addition of nitrate and sulfate reacting to ammonia and hydrogen sulfide, respectively, which are toxic for the animals in higher concentrations, no convincing options have been found to date (*Van Zijderveld et al., 2010*; *Lee & Beauchemin, 2014*).

The first evidence that biochar might act as an electron acceptor and reduce methane production in the rumen came from Vietnam in 2012 (*Leng, Inthapanya & Preston, 2012*; *Leng, Preston & Inthapanya, 2012*). In vitro studies revealed that 0.5% and 1% biochar additions to the ruminal liquid significantly reduced methane production by 10% and 12.7%, respectively. Higher levels of biochar did not further reduce methane production. All experiments were conducted in the presence of 2% urea as a non-protein source of nitrogen. When urea was replaced with nitrate (6% of DM feed intake as $KNO_3$ to supply the same amount of N), methane production decreased by up to 49%.

While both, nitrate and biochar, may act as electron acceptor in the rumen and likely explain at least part of the effect, it is difficult to elucidate on the base of the data provided why the methane reductions by nitrate (−29%) and biochar (−22%) were higher when fed combined (−49%). However, as the effect appears dosage independent (0.5% or 1% biochar) it is unlikely that the two substances reduce methane production by the same mechanisms. It may be hypothesized that the biochar acts as a redox-active electron mediator that takes up electrons from microbial oxidation reactions (e.g., oxidation of acetate to $CO_2$) and donates the electron at a certain distance from the microbial reaction center (at another spot of the same biochar particle) to mediate an abiotic reduction of nitrate (*Saquing, Yu & Chiu, 2016*). Biochar at feeding ratios of about 1% (100 g/day) would not have the capacity to act as terminal electron acceptor for all rumen produced hydrogen considering a daily production of about 200 l methane for the various studies of *Leng, Inthapanya & Preston (2012)* in SE-Asia and up to 500 l methane for typical cattle in Europe or the US. Nitrate (at 6% of DM intake) would have this capacity as terminal electron acceptor but is not efficient as direct electron acceptor in microbial oxidation reaction due to the toxic effects of its reaction products (i.e., nitrite and ammonia).

Another likely mechanism is the biotic reduction of nitrate through Methylomirabilis oxyfera-like bacteria using the supplemented nitrate as an oxygen source for methane oxidation in the rumen. Denitrifying anaerobic methane oxidizing (DAMO) bacteria like *Candidatus* Methylomirabilis oxyfera belonging to the NC10 phylum were shown to efficiently oxidize methane anaerobically in deep lake sediments (*Deutzmann et al., 2014*). NC10 DAMO bacteria were equally found in wetlands (*Shen et al., 2015*), in grassland soils used for animal husbandry (*Bannert et al., 2012*), and with a robust abundance

of $3.8 \times 10^5$ to $6.1 \times 10^6$ copies $g^{-1}$ (dry weight) in flooded paddy fields (*Shen et al., 2014*). DAMO bacteria were further found in the rumen fluid of Xinong Saanen dairy goats in Southern China. The proportion of NC10 in total bacteria in the rumen fluid was 10%, and it could clearly be seen that NC10 mediated nitrate reduction led to reduced enteric methane emissions (*Shen et al., 2016*). Notwithstanding further evidence, it may be hypothesized that the additional effect of combined biochar and nitrate supplements is due to the biotic denitrifying methane oxidation that might further be enhanced through electron accepting and redox mediating properties of the biochar. Systematic investigations to better understand the likely mechanisms are urgently needed.

In vivo experiments showed that methane formation in cattle could be reduced by 20% when 0.6% of biochar was added to the ordinary compound feed (*Leng, Preston & Inthapanya, 2013*). When the same amount of biochar was combined with 6% potassium nitrate, methane emissions decreased by as much as 40%. In addition to reducing methane emissions, highly significant bovine weight gain (+25%) was observed in the experiment as compared to the control, suggesting an increase in feed efficiency and/or reduced energy conversion losses. The biochar in this and the earlier in vitro trial was produced at high temperatures (HTT = 900 °C) from silicon-rich rice husks, which suggests a high electrical conductivity and electron buffering capacity (*Yu et al., 2015*; *Sun et al., 2017*) which may lead to greater efficiency of fodder-decomposing redox reactions. *Leng, Inthapanya & Preston (2013)* have further shown that different biochars have different effects on methane emissions. A likely reason for this are differences in electrical conductivity and in electron buffering (*Sun et al., 2017*) depending on the biomass and pyrolysis temperature, which determine the biochar's properties of transmitting electrons between different bacterial species.

Leng, Inthapanya & Preston also examined the rumen fluid of cattle previously fed with and without biochar. They found that rumen fluid from cows that had been fed biochar produced less methane than rumen fluid from non-biochar-fed cattle. This suggests that the animals fed biochar may have had a different microbial community in the rumen (*Leng, Inthapanya & Preston, 2012*). *Phanthavong et al. (2015)* also found a significant decrease in methane emissions over a 24-h period in in vitro tests with 1% biochar added to a manioc root feed mix, but only by about 7%.

In 2012, a Danish team of researchers led by Hanne Hansen published the results of an in vitro study with large doses of various, but poorly characterized biochars and their effects on methane production of rumen fluids (*Hansen, Storm & Sell, 2012*). All tested biochars (made from wood or straw with slow pyrolysis or gasification) tended ($p = 0.09$) to reduce methane emissions from 11% to 17%, with an activated biochar showing the highest reduction rate. However, the enormously high addition of 9% cannot be considered as viable as this would surely impact feed digestibility on the long term. *Winders et al. (2019)* did not detect any significant reductions on methane emissions in steers over a 23 h period when using the more realistic biochar supplement rates of 0.8% and 3%.

Four biochars (from pine wood chips and corn stover, each pyrolyzed at 350 and 550 °C) were co-fermented at rates of 0.5%, 1%, 2% and 5% in ryegrass silage and used as

feed substrates in an in vitro trial with rumen liquid (*Calvelo Pereira et al., 2014*). None of the biochar treatments revealed any effect on methane production as compared to the control.

Due to the promising results of *Leng, Inthapanya & Preston (2012)* several other research groups have carried out in vitro experiments though without obtaining significant results which, therefore, where not published (Belgium, USA and Germany, Hans-Peter Schmidt, 2018, personal communications). Until today, only the research group of Ron Leng were able to produce and reproduce high reduction rates of methane production both in vitro and in vivo. It is impossible yet to identify a convincing reason or mechanism to explain the strong divergence of the results. It might be due to the particular 900° gasifier rice-husk biochar or to the non-common feed used in their trials (tannin rich cassava roots and foliage that may provide terminal electron acceptors) or the particular rumen microbiota of the South-East Asian cattle that may contain higher rates of DAMO bacteria. The experiments from Europe, New Zealand and America with conventional cattle fodder and standard biochar prudently suggested, that biochar alone (i.e., without nitrate as oxygen source or terminal electron acceptor) may not live up to the expectations to reduce enteric methane emission of cattle (Table 2).

This conclusion is confirmed by a recent and perhaps the most systematic and complete in vitro study to date, at the University of Edinburgh (*Cabeza et al., 2018*). The authors investigated the effects on in vitro rumen gas production and fermentation characteristics of two different rates of biochar (10 and 100 g biochar/kg substrate, i.e., 1% and 10%) made at two different temperatures (HTT 550 or 700 °C) and from five different biomass sources (miscanthus straw, oil seed rape straw, rice husk, soft wood pellets and wheat straw). The methane production was reduced by all biochar treatments and at both concentrations levels by about 5% compared to the control without biochar. There was no significant difference between the different types and amounts of biochar. The absence of significant differences between those very different biochars is puzzling though an important milestone towards the understanding of biochar's mechanisms in animal digestions because there has to be a common cause leading to the same effect between all these different biochars.

A new perspective on the subject was recently put forth by *Saleem et al. (2018)* who used an artificial semi-continuous rumen system to test the effect of a high temperature biochar that was post-pyrolytically treated to acidify the biochar to a pH of 4.8. For a high-forage based diet, 0.5%, 1% and 2% of this acidic biochar reduced methane production by 34%, 16% and 22%, respectively. All other biochars in all of the experiments reviewed here were alkaline (pH between 8 and 11.5). The acidification of biochar not only oxidizes the carbonaceous surfaces and makes the biochar hydrophilic, it also modifies the redox behavior and thus its "affinity" for microbial interaction. As this is, to our knowledge, the first and only experiment to demonstrate a reduction of methane emissions using acidified biochar and as there are no systematic investigations about the acidification effect yet, it is too early to draw a definitive conclusion. However, it is an indication that post-pyrolytic treatment of biochar has the potential to design and optimize the biochar effects in animal digestion, and, notably, to reduce enteric methane emissions.

Table 2 Overview of published studies about biochar effects on enteric methane emissions.

| Daily BC intake/content of rumen liquid | Type of trial | Feedstock | HTT in °C | Activation | Blend | $CH_4$-reduction | Source |
|---|---|---|---|---|---|---|---|
| 0.5% to ruminal liquid | In vitro | Rice husk | 900 | No | 2% urea | 10% | Leng, Inthapanya & Preston (2012) |
| 1% to ruminal liquid | In vitro | Rice husk | 900 | No | 2% urea | 12.7% | Leng, Inthapanya & Preston (2012) |
| 1% to ruminal liquid | In vitro | Rice husk | 900 | No | 6% $KNO_3$ | 49% | Leng, Inthapanya & Preston (2012) |
| 0.6% of feed DM | In vivo | Rice husk | 900 | No | | 20% | Leng, Preston & Inthapanya (2013) |
| 0.6% of feed DM | In vivo | Rice husk | 900 | No | 6% $KNO_3$ | 40% | Leng, Preston & Inthapanya (2013) |
| 1% of feed DM | In vivo | Rice husk | 900 | No | Manioc root feed | 7% | Phanthavong et al. (2015) |
| 9% to ruminal liquid | In vitro | Wood/straw | | Partly | | n.s. (11–17%) | Hansen, Storm & Sell (2012) |
| 1% of DM feed | In vivo | Wood | >600 | | | n.s. | Winders et al. (2019) |
| 0.5%, 1%, 2%, 5% of rumen incubation | In vitro | Wood/corn stover | 350/550 | Ensiled | Mixed to ryegrass before ensiling | n.s. | Calvelo Pereira et al. (2014) |
| 1%, 10% of DM feed | In vitro | Miscanthus straw/ oil seed rape straw/rice husk/ soft wood pellets/ wheat straw | 550/700 | No | | 5% | Cabeza et al. (2018) |
| 0.5%, 1%, 2% of DM feed | In vitro | pine | 400–600 | Acidification to pH 4.8 | | 34%, 16%, 22% | Saleem et al. (2018) |

Note:
The table indicates the reductions of enteric methane emissions of cattle due to biochar feed supplements or additions to rumen liquids summarizing biochar dosages, pyrolysis feedstock and temperature and post-pyrolytic treatments.

The promising results of *Leng, Inthapanya & Preston (2012)* when feeding biochar in combination with nitrate call for systematic investigations of (1) pyrolytic and post pyrolytic treatments (e.g., pyrolysis temperature, activation, acidification), (2) feed blending with terminal electron acceptors (e.g., nitrate, urea and humic substances; *Md Shaiful Islam et al., 2005*), (3) co-feeding with oxygen sources for anaerobic methane oxidation (nitrate) and (4) inoculation with Methylomirabilis oxyfera-like bacteria to oxidize methane.

## Possible side effects of biochar

Based on the literature compiled in the present review, none of the activated and non-activated biochars used as feed additive or veterinary treatment had toxic or negative effects on animals or the environment. No negative side effects were reported either in short-term or long-term administration trials.

There are a growing number of farmers that have been feeding their livestock with biochar additives on a daily basis for several years without noticing negative side-effects (*Kammann et al., 2017*; C. Kammann et al., 2017, personal communications).

However, there are only very few if any long term biochar feeding trials with clinical follow-up (*Struhsaker, Cooney & Siex, 1997*; *Joseph et al., 2015b*). In the absence of clinical long-term feeding trials with biochar, long-term experiments with oral administration of activated carbon to humans seem to indicate rather low risks. The administration of 20–50 g activated biochar daily in uremia patients for 4–20 months did not produce significant side effects (*Yatzidis, 1972*). *Olkkola & Neuvonen (1989)* maintained dosages of 10–20 g administered three times a day over a period of several months in human patients without negative side effects.

The main risks of long-term biochar feeding may arise (1) from shifting microbial species composition in the digestion system (microbiome) and (2) from the potential adsorption of essential feed compounds and/or drugs. Only a few scattered studies have addressed both points.

With regard to the microbiome, the adsorptive capacity of activated biochar for the beneficial bacterial flora in the digestive tract of dairy cows was examined using gram-positive *Enterococcus faecium*, *Bifidobacterium thermophilum* and *Lactobacillus acidophilus* (*Naka et al., 2001*). Although activated biochar certainly adsorbs strains of the normal, healthy bacterial flora too, adsorption of these bacterial strains was significantly lower than the adsorption of the dangerous *E. coli* O157: H7 strain, which is gram-negative. Biochar appeared to positively affect the ratio of (certain) beneficial bacterial flora to (certain) pathogenic flora. However, it must be systematically investigated and mechanistically understood for a much larger number of digestive and pathogenic microorganisms, before a more general conclusion can be drawn. Our review suggests that the impact of biochar on microorganisms depends on the cell envelope, that is, the gram-stain with gram-positive (plasma membrane plus 20–80 nm of peptidoglycan) not being or being less well sorbed to biochar, while gram-negative bacteria (plasma membrane plus 10 nm peptidoglycan plus outer membrane) are better sorbed. However, the structure of the cell envelope and the fact of being gram-positive or negative does not, on its own, indicate whether a bacteria is a pathogen or not.

The potentially selective action of biochars on various bacterial genera opens up the possibility of inoculating the biochar as a carrier matrix with beneficial bacteria, for example, to administer gram-positive *Lactobacilli.* to positively influence the intestinal flora (*Naka et al., 2001*). Different groups of authors have found that pathogens are generally bound more strongly than the native intestinal flora to biochar in the digestive tract (*Naka et al., 2001*; *Watarai, Tana & Koiwa, 2008*; *Choi et al., 2009*; *Chu et al., 2013a*). The hypotheses put forward indicate a possible correlation with more favorable pore size distribution for the adsorption of pathogens, as well as the observation of the (nonspecific) promotion of beneficial microorganisms such as *Lactobacilli*. This combination could positively target the digestive milieu and suppress pathogens.

With regard to sorption, biochar can work against human poisoning and drug overdose (*Park, 1986*), but thus could also counteract intended benefits of drugs. Based on our review, the same can be proclaimed regarding pharmaceuticals used to treat livestock. It is evident that acute, temporary treatment and continuous addition to feed over years do not underlie the same risk assessment. *Fujita et al. (2012)* conducted a comprehensive

study in 2011, where they examined the influence of biochar feeding on hens' health and egg quality. Histopathological studies showed no changes in the digestive tract or in the liver. Examination of the egg yolk showed that fat-soluble vitamins A and D3 did not show a statistically significant trend towards lower concentrations, but that the vitamin E content in the eggs was reduced by about 40% when hens were fed daily with 0.5% biochar (*Fujita et al., 2012*). Although all other quality parameters such as fatty acids, oxidative stability and mineral content in the eggs were not affected by biochar feeding, it was the first evidence that a beneficial compound like a vitamin can be significantly reduced by co-feeding biochar. The above mentioned reduction of carotenoids in egg yolks indicated by changes in yolk color (*Prasai et al., 2018a*) further supports the conclusion that systematic research with well-defined biochars and a focus on liposoluble feed ingredients like vitamin E and carotenoids is needed before industrial scale-up of long-term biochar co-feeding can be safely recommended. However, compared to a large spectrum of other feed additives and ubiquitous pesticide and mycotoxin contamination of animal feed, risks of quality-controlled biochar feed can be considered low, even when supplemented on a regular basis.

## Administration of biochar feed and biochar quality control

Biochar should not be fed without complete biochar analysis and control of all relevant parameters of current feed regulations such as provided by the European Biochar Feed Certificate (*European Biochar Foundation (EBC), 2018*). The analysis should be carried out by an accredited laboratory specialized in biochar and feed analytics. In addition, as required by the EBC, biochar should always be processed and administered moist to avoid the formation of dust (*European Biochar Foundation (EBC), 2012*). If this is respected, biochar can be added to all common feed mixes and is usually mixable with all common feeds. Feed quality biochar may also be added to animal drinking water and, in the case of acute intoxication, activated biochar should be administered in aqueous suspension (*Neuvonen & Olkkola, 1988*). Depending on livestock species, the biochar may also be provided in freely accessible troughs on the pasture or in the stable, without previous mixing into daily feed. Often, the biochar is mixed with popular supplements such as molasses (*Joseph et al., 2015b*) or flavoring such as saccharin, sucrose and the like (*Cooney & Roach, 1979*). Some German and Swiss farmers inject 1% (vol) of biochar into silage towers or silage bales via automated equipment (*O'Toole et al., 2016*).

In many of the experiments cited here, biochar was not administered alone, but in admixture with other functional feed supplements such as humic acid, wood vinegar, sauerkraut juice, eubiotic liquids, stevia, nitrate or tannins, the effect of the mixture often being greater than with separate feeding of the individual components. Those combinations of biochar with various other feed supplements open a huge scope for further research and the reasonable expectation that suitable feed mixtures can be developed for specific purposes and animal species.

The adsorption capacity of biochar depends in particular on the specific surface area, surface charge and the pore size distribution. Activation of biochar significantly increases the specific surface area (from approx. 300 $m^2$ to >900 $m^2$), but the increase in surface

area is mainly due to the opening of micropores (<2 nm). These micropores are mostly too small for the higher molecular weight substances or bacterial pathogens relevant for animal digestion. *Galvano et al. (1996b)* found that biochar with dominating micro porosity (<2 nm) had lower adsorption capacities for mycotoxins due to slow diffusion of these toxins into the pore-system. This was also the case for other investigated toxic compounds such as pesticides, PCBs, dioxins or pathogens, as was demonstrated by *Edrington et al. (1997)* when highly activated biochar did not reduce the toxic effects of aflatoxin in chickens more strongly than non-activated biochar. Therefore, the activation of biochar may not significantly increase the specific adsorption capacity for certain target substances or organisms. To produce a biochar with a particularly high content of accessible meso and macro pores, downstream activation is not necessary and can be achieved merely by adjusting the pyrolysis parameters. Generally speaking, a higher meso-porosity is achieved at pyrolysis temperatures above 600 °C (*Brewer et al., 2014*).

Depending on the activation method, biochar activation and acidification can greatly modify the electron (and proton) mediating capacity (*Chen & McCreery, 1996*), however, to date no systematic research has been done with such modified biochars in animal feeding. Currently, only pyrolysis temperature was identified as main driver for the redox behavior, revealing temperatures between 600 and 800°C as optimal (*Sun et al., 2017*).

To minimize condensate deposition on biochar surfaces and to ensure that PAH contents stay below common thresholds (*European Biochar Foundation (EBC), 2012*) sufficient active degassing of the cooling biochar at the end of the pyrolysis process is mandatory, for example, by using inert gas or by sufficient counter flow ventilation during discharge (*Bucheli, Hilber & Schmidt, 2015*).

Biochars used in the various studies were mainly derived from wood, but also from coconut shells (*Jiya et al., 2013*), rice husk (*Leng, Preston & Inthapanya, 2013*), shea butter stocks (*Ayanwale, Lanko & Kudu, 2006*), bamboo (*Van, Mui & Ledin, 2006*; *Chu et al., 2013a*), corn stover (*Calvelo Pereira et al., 2014*), corncob (*Kana et al., 2011*), straw (*Cabeza et al., 2018*) and many other types of biomass. According to current publications, there is no scientific basis to prefer one source of biomass over another to produce feed-grade biochar. As long as important guidelines for the $H/C_{org}$ ratio (= degree of carbonization), carbon and heavy metal contents, PAHs and other organic pollutants are met, biochar from woody as well as non-woody precursors may safely be used for co-feeding purposes.

The European Biochar Certificate (EBC), a voluntary industry standard, has been controlling and certifying the quality of biochar for use in animal feed since January 2016 (*European Biochar Foundation (EBC), 2018*). To date, six biochar producing companies have obtained the EBC-feed certificate (*European Biochar Foundation (EBC), 2013*). The EBC Feed Certificate guarantees compliance with all feed limits prescribed by the EU regulations and, moreover, certifies sustainable, climate friendly production (*European Biochar Foundation (EBC), 2018*).

## CONCLUSIONS

The use of biochar as a feed additive has the potential to improve animal health, feed efficiency and livestock productivity, to reduce nutrient losses and greenhouse gas

emissions and to increase manure quality and thus soil fertility. In combination with other good farmer practices, biochar could improve the overall sustainability of animal husbandry. The analysis of 112 scientific papers on biochar feed supplements has shown that in most studies and for all farm animal species, positive effects on different parameters such as growth, digestion, feed efficiency, toxin adsorption, blood levels, meat quality and/or emissions could be found. However, a relevant part of the studies obtained results that were not statistically significant. Most importantly, no significant negative effects on animal health were found in any of the reviewed publications.

It is undeniable that, despite the large number of scientific publications, further research is urgently needed to unravel the mechanisms underlying the observed results and to optimize biochar-based feed products. This applies in particular to the characterization of the biochar itself, which in the majority of studies was insufficiently analyzed. The electrochemical interaction of biochar and organic systems is extremely complex and needs considerable more fundamental research and systematic in vivo trials. Moreover, if biochar's role within animal digestion is mainly to act as a mediator and carrier substance, the combination with other feed additives and inoculants may be mandatory to achieve the full functionality of biochar for its beneficial use in animal digestion and animal health.

Based on the scientific literature published so far, it can be concluded that (1) a general efficacy of biochar as feed supplement can be observed and (2) biochar feeding can be considered safe at least for feeding periods of several months. Despite this positive assessment, regular feeding of biochar should never induce livestock farmers to compromise on the quality of feed and animal welfare standards.

### Funding
This study was financed by the BioC project of the r4d call of the Swiss National Science Foundation. Claudia Kammann received financial support from the BMBF-funded project BioCAP-CCS, grants no. 01LS1620A and 01LS1620B. The funders had no role in study design, data collection and analysis, decision to publish, or preparation of the manuscript.

### Grant Disclosures
The following grant information was disclosed by the authors:
BioC project of the r4d call of the Swiss National Science Foundation.
BMBF-funded project BioCAP-CCS: 01LS1620A and 01LS1620B.

### Competing Interests
The authors declare that they have no competing interests. Hans-Peter Schmidt and Nikolas Hagemann are employed by Carbon Strategies, Ithaka Institute. Nikolas Hagemann is employed by Agroscope. Kathleen Draper is the director of Ithaka Institute for Carbon Intelligence.

## Author Contributions

- Hans-Peter Schmidt conceived and designed the experiments, analyzed the data, prepared figures and/or tables, authored or reviewed drafts of the paper, approved the final draft.
- Nikolas Hagemann analyzed the data, authored or reviewed drafts of the paper, approved the final draft.
- Kathleen Draper analyzed the data, authored or reviewed drafts of the paper, approved the final draft.
- Claudia Kammann conceived and designed the experiments, analyzed the data, authored or reviewed drafts of the paper, approved the final draft.

## Data Availability

There were no raw data used in this literature review.

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
