# Peer review of "The use of biochar in animal feeding"

_PeerJ, doi:10.7717/peerj.7373_

## Round 0.1 · original submission · Major Revisions

Apologies that the review took so long, thank you for your patience. Finding reviewers for long pieces is challenging nowadays. Both reviewers (as well as myself) really appreciated the work. Please consider carefully all the suggestions for improvements.

Reviewer 1 ·

Basic reporting

This is an excellent paper with a lot of data.

It would be much improved if further summary tables were included. I would suggest a summary table for each of the animals and highlight whether the biochars were activated with acids or urea/nitrates/wood vinegar lactic acids added. Maybe also a separate table that summarises methane production

Experimental design

No Comment

Validity of the findings

No Comment

Additional comments

Congratulations. This paper has been required for a long time

Reviewer 2 ·

Basic reporting

Well written and comprehensive. Appreciate authors’ effort to evaluate all trials involving biochar. I am not aware of another review (at least in English) that covers this subject.

Experimental design

The search seemed fair and complete. Research results that I am familiar with were all included. A few suggestions on specific references are given below in the general comments. Discussion was organized into logical sections which made reading enjoyable.

Validity of the findings

Written with apparent impartiality and an effort to discuss all aspects of biochar relative to animal feeding. Several places where I intended to ask for further explanation or evidence, but it was then provided in the subsequent paragraph. Appreciate all of the useful suggestions for further research to improve our understanding and potential uses of biochar

Additional comments

Line 29: it is not clear to me what is meant by ‘blood levels’
L62: large yield increases in what crops?
L79: was this based on anecdotal observations or research data?
L81: please define CEC
L429: I don’t believe the Erickson, 2011 article measured or addressed cow health. This was a short term study and not intended to quantify health effects.
In the US one of FDA’s concerns is that biochar may mask disease symptoms preventing treatment of the underlying cause of disease. This is prevalent throughout chapter 3 discussing potential adsorption of toxic substances, when it would be far better to instead limit or eradicate the feeding of these toxic substances.
L609: the link to botulism is suspected, as you write, but is still very much a theory. It is also very specific to animals that have been fed high levels of glyphosate, which I believe is rare. Recommend expanding discussion very briefly.
L642: Are these compounds common in animal feed at levels that cause a concern? Important to also discuss that because these compounds can be bound by biochar, it is critical that we not feed ‘contaminated’ biochar to animals-whether this contamination comes through the production process or by absorption from the environment after production.
L755: I am concerned whether these were otherwise healthy cows or cows with clinical disease(s)?
L756: Leng references---I believe Lines 1778 and 1782 are redundant. The reference I am most familiar with (Leng et al., 2012) studied biochar inclusion in diets of both male and female cattle, thus the word ‘cows’ is not appropriate. (Also in the table)
L792: would these nutrients have been excreted on the pasture even without the addition of biochar to the diet? I am not following how biochar changes this process-does it make the nutrients more plant available, make the nutrients more stable once excreted, remove more nutrients from the digestive tract than would normally be excreted (and if so how would that impact animal performance), or some other mechanism? My simple understanding would be that nutrients are made more stable—less N volatilized and less P leached into groundwater—is this correct?
L803-over what time frame were these measurements made? Long term or short term study?
L804-809: what type of silage (corn, wheat, grass…)?
L810-814: this paragraph is not overly well developed. Recommended removing or adding discussion to make it more clear what in vitro measurements were made.
L822: what about feed efficiency (gain:feed)? With a greater CP intake I assume this is coming from greater overall intake of feed.
L861: recommend stating as ‘without the negative side effects to the environment that antibiotics can have’
L982: reword sentence-not clear as written
L1016-1017: should this be ‘NH3 emissions deceased by 31%’?
L1022: recommend rewording to demonstrate that considerable harm can be done if effluents are not handled correctly. Nutrient rich effluent can be a resource if used correctly.
L1025: in the aquaculture studies was the biochar fed as part of a complete feed or offered separately from the feed?
L1055: recommend removing first sentence of this paragraph as the issue is much too complex to reduce to 1 sentence. An entire review could (and has) been written on sources of GHG emissions and interactions with climate change-which seems outside the scope of this review. The rest of the paragraph sufficiently indicates the importance of reducing CH4 emissions.
L1070: almost entirely through eructation (Murray, Bryant, and Leng, 1976; Br. J. Nutr. 36:1)
L1080: fat has been another feed source studied, biohydrogenation of fatty acids in the rumen utilizes H
L1099: daily production of methane from cattle is largely feed intake driven, larger animals consume more and thus have larger emission of methane. The cattle used in the Leng study were quite small (80 to 100 kg) compared to typical animals in the US or Europe (>200 kg at weaning). 500 L of methane daily would be for larger (mature) animals
L1185: it is my understanding that the Erickson et al. (2011) article that you reference in lines 425-433 also used an acidified biochar product (acid-washed activated carbon product made from lignite coal)
L1261: it would be very useful to have a list of parameters that should be analyzed as a recommendation to future researchers. As you point out previously, veterinarians and nutritionists are not steeped in biochar knowledge or vocabulary and a list of important aspects would be very helpful.
L1525: I attempted to look at the EBC, 2012 reference to see if the above information was contained there, but the link is not active/working—however the article was easy to find. Is this the list of parameters that you would recommend?
L2053: I believe the Winders et al. 2018 article has been published as a complete article (Translational Animal Science, 2019 https://doi.org/10.1093/tas/txz027)
A very important point is the need for safe and more uniform biochar being produced if feeding it to livestock in order to prevent unintentional poisoning—appreciate how you address this in the final paragraphs.

---

## Round 0.2 · accepted · Accept

Congratulations! This is a comprehensive and very timely review!